



# Quantification of methane emissions from hotspots and during COVID-19 using a global atmospheric inversion

Joe McNorton[1], Nicolas Bousserez[1], Anna Agustí-Panareda[1], Gianpaolo Balsamo[1], Richard Engelen[1], Vincent Huijnen[2], Antje Inness[1], Zak Kipling[1], Mark Parrington[1], Roberto Ribas[1]

[1]European Centre for Medium-Range Weather Forecasts, Reading, RG2 9AX, UK
[2]Royal Netherlands Meteorological Institute (KNMI), De Bilt, NL-3731, Netherlands

*Correspondence to*: Joe McNorton (joe.mcnorton@ecmwf.int)

**Abstract.**

Concentrations of atmospheric methane ($CH_4$), the second most important greenhouse gas, continue to grow. In recent years this growth rate has increased further (2020: +14.7 ppb), the cause of which remains largely unknown. Here, we demonstrate a high-resolution (~80km), short-window (24-hour) 4D-Var global inversion system based on the ECMWF Integrated Forecasting System (IFS) and newly available satellite observations. The largest national disagreement found between prior (63.1 Tg yr-1) and posterior (59.8 Tg yr-1) $CH_4$ emissions is from China, mainly attributed to the energy sector. Emissions estimated form our global system agree well with previous basin-wide regional studies and point source specific studies. Emission events (leaks/blowouts) >10 t hr-1 were detected, but without accurate prior uncertainty information, were not well quantified. Our results suggest that global anthropogenic $CH_4$ emissions for 2020 were 5.7 Tg yr-1 (+1.6%) higher than for 2019, mainly attributed to the energy and agricultural sectors. Regionally, the largest 2020 increases were seen from China (+2.6 Tg yr-1, 4.3%), with smaller increases from India (+0.8 Tg yr-1, 2.2%) and Indonesia (+0.3 Tg yr-1, 2.6%). Results show the rise in emissions, and subsequent atmospheric growth, would have occurred with or without the COVID-19 slowdown. During the onset of the global slowdown (March-April, 2020) energy sector $CH_4$ emissions from China increased; however, during later months (May-June, 2020) emissions decreased below expected pre-slowdown levels. The accumulated impact of the slowdown on $CH_4$ emissions from March-June 2020 is found to be small. Changes in atmospheric chemistry, not investigated here, may have contributed to the observed growth in 2020. Future work aims to develop the global IFS inversion system and to extend the 4D-Var window-length using a hybrid ensemble-variational method.

## 1 Introduction

Atmospheric methane as a long-lived greenhouse gas (GHG) has contributed to ~23% of the additional radiative forcing since 1750 (Etminan et al., 2016), second only to $CO_2$. Near-surface concentrations have more than doubled since the pre-industrial era, with the global average dry air mole fraction reaching 1891 ppb in 2020 (gml.noaa.gov, 2021). This growth





can mainly be attributed to increased anthropogenic emissions from agriculture, biomass burning, fossil fuel extraction and use, and waste (Etheridge et al., 1998).

35 The reduction in global human activities, triggered by the COVID-19 pandemic, provided an opportunity to assess the impact of potential rapid climate mitigation strategies to reduce GHG emissions (Diffenbaugh et al., 2020). The sectors most obviously affected by the slowdown, e.g., transport and industry, are directly associated with fluxes of short-lived pollutants (Ming et al., 2020) and CO2 (Le Quéré et al., 2020), and less so CH4. The change in energy and fuel demand is estimated to have reduced oil and gas CH4 emissions by 10 % for 2020 when compared to 2019 (IEA, 2021). Similarly, a recent study 40 found reduced emissions from the largest oil producing basin in the USA, the Permian Basin, between April and May of 2020 (Lyon et al., 2020). Despite this, during 2020 atmospheric concentrations of CH4 grew by 14.7 ppb, the largest amount since records began in the early 1980s (NOAA, 2021). The reduction in demand could have increased venting/flaring when extracting fossil fuels, leading to increased atmospheric concentrations. The remaining CH4 source sectors were not expected to have been noticeably impacted by changes in activity during the slowdown. The reduced emissions of OH-45 forming nitrogen oxides (NOx) during the slowdown may have reduced the CH4 sink (Stevenson et al., 2021), however another recent study suggests this impact may only be small (Weber et al., 2020).

Given the relatively large atmospheric variability of CH4 concentrations and accurate measurements, the quantification and attribution of emissions is possible using inverse modelling based on both in-situ (e.g. Wilson et al., 2016; McNorton et al., 50 2018) and satellite observations (e.g. Bergamaschi et al., 2018; Maasakkers et al., 2019). Global atmospheric flux inversions (e.g. Segers and Houweling, 2018; Qu et al., 2021) are typically performed at a coarse spatiotemporal resolution (~monthly, >1°), for which localised events (e.g. leaks and blowouts) are difficult to detect. Additionally, previous attempts to quantify emissions have been restricted by limited surface and satellite observations. The Scanning Imaging Absorption spectrometer for Atmospheric CartograpHY (SCIAMACHY) provided the first total column CH4 (XCH4) measurements from space. 55 These observations were superseded by the Greenhouse gases Observing SATellite (GOSAT) in 2009, offering higher sensitivity and spatial resolution (~10 km). GOSAT is limited by a relatively narrow spatial sampling restricting the coverage. Both instruments have been used to constrain CH4 surface fluxes in inversion studies (e.g. Frankenberg et al., 2005; Maasakkers et al.., 2019). The TROPOspheric Monitoring Instrument (TROPOMI) instrument on-board Sentinel-5P, launched in 2017, provides global high-resolution (~7 km) XCH4 observations with an improved spatiotemporal coverage 60 and precision (Veefkind et al., 2012; Hu et al., 2018). These newly available observations provide the opportunity to detect CH4 hotspots (Barré et al., 2020) and potentially constrain CH4 fluxes at high spatiotemporal resolution (Pandey et al., 2019; Zhang et al., 2020).

This study presents and evaluates a first version of the new capabilities introduced in the ECMWF Integrated Forecasting 65 System (IFS) to estimate emissions of greenhouse gases and atmospheric pollutants using satellite observations of their





atmospheric concentrations. The system is being developed in the framework of the EU-funded Copernicus CO2 project (coco2-project.eu, 2021) and its precursor, the CO2 Human Emission project (Balsamo et al., 2021) as the global prototype for a new Copernicus anthropogenic CO2 emissions monitoring and verification support capacity (Janssens-Maenhout et al., 2020). For this paper, the focus on CH4 emissions allows to benefit from greater observability from remote-sensing

(compared to CO2) and suitably large spatiotemporal variability, addressing three main outstanding questions. First, are CH4 emission hotspots quantifiable using multiple sensors and a high-resolution global short-window 4D-Var system when accounting for meteorological errors? Second, how well do concentrations generated using posterior emission estimates agree with independent observations and existing studies? Third, is the system capable of assessing potential longer-term trends during the COVID-19 pandemic slowdown?


The following sections, 2.1 and 2.2, outline model methodology, detailing the 4D-Var inversion system used and prior assumptions made. Section 2.3 describes the observations assimilated into the inversion system. Section 3.1 identifies suitable prior uncertainty assumptions in CH4 fluxes. Section 3.2 provides a global overview of posterior fluxes and the relative changes from prior estimates. Section 3.3 evaluates the system using a range of regional and persistent point source

case studies. Section 3.4 Investigates the feasibility to quantify emissions at a high spatial and temporal resolution using case studies. Section 3.5 investigates the influence of the global slowdown triggered by the COVID-19 pandemic on CH4 emissions. Section 4 discusses the findings and relevance to the wider community including limitations and suggestions for future work.

## 2. Methods

### 2.1 Forward model

The ECMWF global Integrated Forecasting System (IFS), which provides the operational Copernicus Atmosphere Monitoring Service (CAMS, https://atmosphere.copernicus.eu/) greenhouse gas (GHG) forecast (Agusti-Panareda et al., 2019), was used to generate the forward model integrations used in this study. These were performed from January to June of 2019 and January to September of 2020, with additional case study simulations performed for June 2018 and November

2019. Simulations were performed using a horizontal cubic octahedral reduced Gaussian grid (TCo399: ~25km) and 137 vertical levels with coupled meteorology at operational forecast timesteps of 15 minutes and 3 hourly output.

Monthly gridded prior estimates of anthropogenic emissions were taken from the CAMS global emissions product, CAMS-GLOB-ANT v4.2, (Granier et al., 2019), which combines existing products (e.g. EDGAR: Cippa et al., 2018; CEDS: Hoesly

et al., 2018). The Global Fire Assimilation System (GFAS) provided daily biomass burning emissions (Kaiser et al., 2012). We used a monthly climatology of wetland emissions based on the LPJ-WHyMe model (Spahni et al., 2011). Remaining



fluxes from oceans (Lambert and Schmidt, 1993; Houweling et al., 1999), termites (Sanderson, 1996) and wild animals (Houweling et al., 1999) were used at the highest available spatiotemporal resolution.

The atmospheric CH4 sink comprised of a monthly mean climatological loss rate field (Bergamaschi et al., 2009), which represents loss reactions with hydroxyl, chlorine and atomic oxygen radicals. A gridded surface soil sink was also used (Ridgwell et al., 1999). Initial conditions for the 3D atmospheric state of CH4 were taken from the CAMS CH4 inversion product (Segers and Houweling, 2018).

## 2.2 Inverse Model

### 2.2.1 4D-Variational inversion

We used the 4D-Var IFS system, cycle 47R1 used operationally at ECMWF between June 2020 and May 2021. More detailed information on the IFS 4D-Var system can be found in Rabier et al. (2000) and Courtier et al. (1994). The incremental algorithm used consists of solving a series a quadratic minimisation problems (inner-loop) constructed by linearising the initial (non-linear) cost function around updated estimates of the state vector (outer-loop). To constrain
surface emissions, the state vector is augmented by a parameter control vector that consists of a 2D scaling factor applied to a prior emission inventory (see Sec. 2.2.2), based on Massart et al. (2021). In our configuration, the posterior scaling factors are optimised on a regular 2D grid (~80 km) within a 24-hour window and then applied to the prior emission inventory defined on a grid of ~10 km resolution (Figure 1). Prior emission errors are assumed to be independent between 24-hour inversion cycles (i.e., each 24-hour inversion uses the same uniform scaling factor of 1 and the same prior errors). This
choice was driven by the lack of information about temporal error correlations in current prior inventories. Posterior errors in methane emissions and 3D state are not propagated forward across data assimilation cycles in this configuration, which is a technical limitation of our current system and will be addressed in subsequent versions. We use an online 4D-Var data assimilation system, where the meteorological fields are part of the control vector and optimised jointly with the emission scaling factors. As a result, the transport errors associated with uncertainties in the initial conditions of the meteorological
variables are accounted for in our inversion. This is in contrast with widely used offline inversion systems, wherein transport error are typically prescribed on an ad-hoc basis and fixed. Note that in our experiments the background errors for the meteorological variables at initial time are constructed based on a climatology, and therefore are not flow-dependent.

The scaling factors derived from the inversion were applied to sector specific prior maps for source attribution. A caveat to
this approach is the assumption that collocated sectors have the same scaling factor applied, which can only be overcome with the use of co-emitted species observations such as ethane or isotopologues (e.g. McNorton et al., 2018). However, this is unlikely to noticeably impact these results as at the relatively high increment resolution used (~80km) CH4 sectors are rarely collocated. Missing sources in the prior are also not accounted for when using a posterior scaling factor.





### 2.2.2 Prior information

Anthropogenic sector specific grid cell uncertainties, taken from Maasakkers et al. (2016), provided the initial prior estimate for countries with well-developed statistical infrastructures or Annex I countries (IPCC, 2006). For Non-Annex I countries, the same sector specific uncertainties were further increased by 50%. Globally, constant wetland uncertainties were estimated at 58%, taken as the standard deviation from the WetCHARTs ensemble (Bloom et al., 2017). Initially, all other biogenic uncertainties were estimated as 100%. The atmospheric sink was not optimised by the inversion. Sensitivity

experiments where prior errors were perturbed and validated against independent observations were used to evaluate prior uncertainty assumptions (supplementary table 1). Given anthropogenic emissions are typically from point sources (e.g. fossil fuel extraction), we assumed no spatial prior error correlation given the derived increments are at a ~80 km. Wetland emissions would typically require defined spatial correlations, however given the uncertainty of these structures, the focus of this study being anthropogenic emissions and co-located emissions from wetland and anthropogenic sources we have chosen

to omit these for simplicity. Total grid cell uncertainties, used in the control vector, were calculated with the error propagation method. All prior uncertainties are assumed to have a log-normal distribution to prevent negative emissions.

### 2.3 Observations

The observations used in the meteorological component of the IFS 4D-Var system include satellite radiances, conventional ground based and radiosondes, and aircrafts and ships data, for which the coverage and quality is constantly monitored prior

the assimilation. With specific focus on CH4, the TROPOMI instrument on-board the Sentinel-5 Precursor satellite provides near-global daily coverage of XCH4 with a nadir ground pixel size of 7 km x 7 km and near-surface sensitivity (Veefkind et al., 2012; Lorente et al., 2021). We used operational observations, which became available in April 2018 and were bias corrected, as in Barré et al. (2020). An example representation of daily satellite coverage, which is applicable within a 24-hour 4D-Var window, is shown in Supplementary figure 1. TROPOMI uncertainties provided as part of the CH4 product

were applied within the minimisation routine and averaging kernels were used. Additional CH4 observations from the Infrared Atmospheric Sounding Interferometer (IASI) and GOSAT are assimilated into the system to provide additional constraints as described by Massart et al., (2014).

### 3. Results

Several simulations were performed. First, a suite of sensitivity experiments was performed to identify an appropriate prior

flux uncertainty (section 3.1). This was then used to investigate global emissions (section 3.2), specific emission events (section 3.3 and 3.4) and perform comparative source attribution of $CH_4$ fluxes during the COVID-19 global slowdown (section 3.5). A full list of simulations is provided in supplementary table 1. Between mid- to late-March 2020 most countries implemented slowdown measures, which reduced socioeconomic activities (Hale et al., 2021). These measures typically lasted until May or June when certain activities were progressively reintroduced, although not to pre-slowdown



levels. To investigate the impact of these measures on $CH_4$ emissions, relative to previous years, we perform simulations from January, when slowdown restrictions were limited to China, to June for 2019 and 2020.

### 3.1 Evaluation

To assess the suitability of our prescribed prior error in $CH_4$ emissions, 6 sensitivity inversions with a range of uncertainties were performed (see supplementary table 1). We also performed an additional experiment where only the initial 3D
atmospheric concentration of $CH_4$ was optimised. Optimised emissions were then used in forward model simulations, which were evaluated against $XCH_4$ measurements from 16 Total Column Carbon Observing Network (TCCON) sites (Wunch et al., 2011). TCCON averaging kernels were applied to model profiles as described in Massart et al. (2016). Results show improved performance when including flux scaling factors in the control vector when compared to only optimizing the initial 3D-state (Supplementary figure 2). When evaluating $XCH_4$ concentrations simulated with optimised emissions, the all-site
average lowest standard error (6.8 ppb), absolute mean bias (7.52 ppb) and highest R-value (0.74) was found for the mapped prior error described in section 2.2.2. All subsequent experiments used the mapped prior uncertainty, typically ranging from 50-150%.

### 3.2 Global Emission Estimates

As human activities have changed in 2020 in response to the COVID-19 pandemic we first investigated the difference
between prior and posterior emissions for a business as usual year, 2019. Emissions were estimated using the 4D-Var global inversion system described in Section 2.2 from January to June 2019. The resulting fire and wetland emissions are likely to be an inaccurate estimate of annual emissions because of the strong seasonality of both sources. TROPOMI observations do not provide full global coverage within our 24-hour 4D-Var window, resulting in emissions not being constrained over large areas. To produce meaningful spatiotemporal budgets of posterior emissions the posterior error covariance should be
accounted for. Because this latter quantity is currently lacking in our system, we chose to compute posterior emission budgets based on a subset of grid cells that are significantly constrained by the observations. To this aim, in our analysis grid cells whose distance to an observation were greater than 1° were discarded. For each selected grid cell, we apply the monthly mean posterior scaling factor to our prior emission inventory to provide a posterior emission estimate. Globally, we found total posterior emission estimates (528.2 Tg yr$^{-1}$) for 2019 were 4.7 Tg yr$^{-1}$ smaller than prior estimates (532.9 Tg yr$^{-1}$).
Within national boundaries, both negative and positive adjustments in emissions often occur (Figure 2b). Moreover, we found that when averaged over the 6-month period, considerable changes, relative to the prior, are from anthropogenic sources (+4.7 Tg yr$^{-1}$).

At national scales, anthropogenic emission differences between the prior (63.1 Tg yr$^{-1}$) and the posterior (59.8 Tg yr$^{-1}$) were
found to be largest over China (Figure 2c). The potential overestimation in bottom-up emission estimates from China is well documented (e.g. Cheewaphongphan et al., 2019), although the magnitude of this overestimation is uncertain. Using prior





emission maps, we distributed total posterior emissions into 6 sector specific categories; energy, agriculture, waste, other anthropogenic, wetlands and fires. In agreement with multiple inverse studies (Deng et al., 2021) most of the overestimated emissions from China are found to originate from the energy sector (1.9 Tg yr$^{-1}$) and specifically from the coal mining

regions of Outer Mongolia, Shaanxi and Shanxi. Relative to the prior, posterior emissions are reduced from India (-3.0%) and Pakistan (-1.1%), increased from Brazil (+1.3%) and less than 1% different for the USA (0.5%), Indonesia (0.3%), EU27+UK (+0.1%) and Russia (-0.7%). Except for Russia and Indonesia, these bring emission estimates in closer agreement with other top-down studies (e.g. Deng et al., 2021).

### 3.3 Emission estimates for Regions and Point Sources

The feasibility to detect and quantify emission hotspots on a global scale using a relatively high resolution increment grid (~80 km, daily), a high resolution prior emission grid (~9km, monthly) and multi-sensor data was evaluated using previously documented case studies (e.g. Zhang et al., 2020 Varon et al., 2020). Preliminary work by Barre et al., 2020 combined high-resolution IFS forecasts (~9 km) with TROPOMI observations to detect missing emission sources based on a statistical analysis; here we attempted to extend this to the quantification of emissions in a robust atmospheric transport inversion

framework. To filter posterior estimates which provided little or no updated information we omitted daily grid cells associated with poor observation constraints (see supplement figure 1). Future developments will account for posterior error reduction in our analysis. Efforts are ongoing to include an ensemble-based estimate of the posterior emission errors in our system to provide a more robust evaluation. Posterior emissions and comparisons with existing studies for several case studies are provided in table 1.

### 210 3.3.1 Regional emissions - Permian Basin, USA

The Permian Basin, an area of ~400km$^2$, is the largest oil-producing basin in the USA. Previous studies identified an underestimation in inventory estimates of CH$_4$ fluxes in this region (Alvarez et al., 2018; Robertson et al., 2020; Zhang et al., 2020). In recent years oil production in the basin has undergone rapid expansion with output of crude oil quadrupling and natural gas more than doubling between 2007 and 2018 (Zhang et al., 2020). Given the rapid expansion and the lag in uptake

of statistical information to inform the prior inventory, it is likely that the prior used here underestimates emissions from the region. Variability in atmospheric transport over the basin noticeably impacts observed XCH$_4$ enhancements (Crosman et al., 2021), therefore an accurate high-resolution representation of transport is required to quantify emissions. The IFS system, used here, is suitable to address such a problem as it performs an online assimilation of atmospheric composition and meteorological observations therefore providing an improved representation of transport uncertainty.


Using only dates when nearby TROPOMI observations were available (237/485), inversions for the 15 months available (January to June 2019 and January to September 2020) provided average posterior emissions of 2.3±0.5 Tg yr$^{-1}$ over the 6°x4° domain, centred around 32°N, 103°W (Figure 3). The standard deviation value indicates the daily variability and not





the posterior uncertainty. This is a considerable increase from the prior $2.0\pm0.0$ Tg yr$^{-1}$. The estimated flux brings emissions

closer to, but remains lower than, a recent 4D-Var inversion estimate, $2.9\pm0.5$ Tg yr$^{-1}$ (Zhang et al., 2020). A small positive trend is identified over the basin ($+150$ kt yr$^{-1}$). While it is difficult to diagnose the cause of the difference in posterior estimates, one possibility is the larger prior uncertainty used in Zhang et al. (2020). Additionally, transport uncertainties associated with initial meteorological conditions are accounted for in our online inversion system, which might significantly impact the derived emissions.

### 3.3.2 Regional emissions - Bakken Formation, USA/Canada

The Bakken Formation, predominantly in North Dakota, is a major oil-producing region both within the USA and Canada. The rig count in the region has declined in recent years; however, except for during the initial 2020 global slowdown, both oil and gas production have seen large increases in the past decade (EIA, 2021). During recent years various management methods have sought to reduce fugitive emissions from the region, however it remains one of the largest emitting regions

within North America (Schneising et al., 2020).

A previous study estimated average $CH_4$ emissions from the Bakken Formation between 2018 and 2019 of $0.89\pm0.56$ Tg yr$^{-1}$ (Schneising et al., 2020). These were estimated using a Gaussian integral method and TROPOMI data. Our prior emissions ($1.03$ Tg yr$^{-1}$) for a $1°\times1°$ domain centred around $48.5°N$, $103°W$ for 2019 are larger than those previous derived estimates.

Our posterior results for 2019 ($0.93\pm0.48$ Tg yr$^{-1}$) show a variable but positive trend in emissions from the region (Figure 4). These estimates agree with those derived by Schneising et al. (2020). For 2020, a period not included in their study, we find larger average emissions relative to 2019 ($1.03\pm0.63$ Tg yr$^{-1}$). Unlike for the Permian Basin example, the agreement found here is based upon two different top-down approaches, our 4D-Var IFS system and the Gaussian integral method of Schneising et al., (2020).


A possible $CH_4$ emission event is observed on the 4th September 2020 where emissions were estimated to increase by 350% from a 2020 average of 120 t hr$^{-1}$ to 410 t hr$^{-1}$, which over the 24-hour period equates to an additional 7 kt $CH_4$. The source of this previously undocumented event is not clear, an incident reported at The Steelman Gas Plant in Saskatchewan, Canada is a possibility; however, accurate attribution requires further investigation (Saskatchewan.ca, 2021). Several similar events

of slightly smaller magnitude are also observed, the causes of these require further investigation.

### 3.3.3 Regional natural emissions - Lake Chad, Africa

The hydrology of Lake Chad and the surrounding area has recently undergone substantial variability on timescales ranging from seasonal to decadal (Pham-Duc et al., 2020), which is expected to have impacted both natural and anthropogenic emissions in the region. A recent study, using a similar prior to the one used here, performed a top-down inversion over

tropical Africa using GEOS-Chem and GOSAT observations and found posterior emissions increased relative to their prior



over Lake Chad between 2016 and 2018, although these are not quantified (Figure 3c of Lunt et al., 2019). Our results for 2019 and 2020 for a 1°x1° box centred around the lake (13.0°N, 14.3°E) show posterior emissions (0.38±0.05 Tg yr⁻¹ are 11% higher than prior emissions (0.35±0.02 Tg yr⁻¹) (Figure 5). Observations are only available over the region for 65 out of 485 days, making estimations of the seasonal shift between the posterior and prior difficult. We are unable to attribute the increased emissions to a specific sector; however, based on prior information, it is likely to be from agricultural livestock or wetland sources. If this region-wide increment is the result of wetland emissions, with further refinement and accurate characterisation of prior error correlations, our system could be used to quantify emissions over wetland regions.

### 3.3.4 Point source emissions - Appin Colliery, Australia)

The Appin Colliery (34.2°S, 150.8°E), in New South Wales, Australia is an underground coal mine previously noted for having high $CH_4$ emissions (Varon et al., 2020). It represents a single point source, which is challenging to quantify as there are several nearby emission sources including landfills, dairy facilities, and a gas processing plant. Varon et al., (2020) used the high-resolution GHGSat-D instrument and, integrated mass enhancement (IME) and cross-sectional flux (CSF) methods calibrated with large eddy simulations to derive vent emissions from the mine between 2016 and 2018. They estimated mean $CH_4$ emissions of 5.9 t hr⁻¹ (IME) and 5.0 t hr⁻¹ (CSF), lower than the prior used here (6.7±0.1 t hr⁻¹, fugitive only: 6.0±0.1 t hr⁻¹). We derived 2019-2020 average grid cell emissions of 6.4±0.7 t hr⁻¹. Assuming little or no change in emissions between their 2016-2018 study period and our 2019-2020 estimate, our derived, fugitive-only, emissions (5.7±0.6 t hr⁻¹) agree well with their findings (Figure 6). For 2019, a business as usual year, which is nearer to the time period investigated in their study, fugitive emissions are even lower (5.3±0.7 t hr⁻¹). These results suggest our inversion is capable of detecting biases in the prior from point sources, given sufficient observations (100/485 days observed), a relatively large point source (>~5 t hr⁻¹) and a suitable prior uncertainty estimate. Prior emission estimates appear to be in better agreement with our posterior in 2020, suggesting an increase in emissions, most likely from the Colliery given it is the dominant source in the region.

### 3.4 Emission estimates for Temporary and Shifting Sources

The following 4 cases assess the quantification of emissions from specific release events, step changes in emissions or short-term observation periods, using documented examples and previously unexplored sources. As with the regional comparisons in the previous section, evaluation of the system is performed against multiple emission estimation systems beyond the 4D-Var approach used here.

### 3.4.1 Feasibility of estimating blow-out emissions - Eagle Ford Blowout, USA (November 2019)

On 1ˢᵗ November 2019, a blowout event occurred at a gas well in the Eagle Ford Shale in Texas (28.9°N, 97.6°W), which was followed by a diminishing 20-day release event (Cusworth et al., 2021). Cusworth et al. (2021) estimated emissions of the blowout using several estimation techniques, including the Integrated Methane Enhancement algorithm (Varon et al.,





2018), and multiple observation platforms, including TROPOMI. Observations directly over the blowout were made from TROPOMI on the 2nd, 3rd, 15th and 18th of November 2019. We further extended our analysis to all observations made between 15th October and 28th November 2019 within 2°x2° domain centred around the blowout (Figure 7). We found when

blowout emissions peaked on the 1st/2nd November 2019, posterior emissions at the site were ~40% higher than prior emissions; however, the magnitude of the posterior emissions (2.5 t hr$^{-1}$) is noticeably lower than the 28-61 t hr$^{-1}$ previously estimated (Cusworth et al., 2021). As expected, posterior emission estimates return to near prior levels after the initial blowout (Figure 7c-e). Estimates provide by Cusworth et al., (2021) would require more than a 1,500% increase in emissions relative to our prior which is unlikely to be achieved with our relatively modest prior error (87%). It is likely given the model

resolution and prior information that posterior emissions are incorrectly attributed to nearby grid cells. This is evident in the mapped scaling factors, which show increases incorrectly applied slightly to the west of the blowout location. Within a 4°x4° domain surrounding the blowout site posterior and prior emissions typically agree well for months excluding November, suggesting any differences occurring in November, could be attributed to the well blowout. Based on this assumption we used the residual from the posterior minus the prior to estimate blowout emissions on the 2nd November 2019 of 140 t hr$^{-1}$,

which is more than double the estimate of Cusworth et al. (2021). These results suggest that the system, as presented here, can detect such events but cannot accurately quantify a well blowout of this magnitude over an oil field.  It could however be used as a crude quantification of emissions from such a blowout over a larger domain, assuming other sources are well known. A more accurate quantification of emissions from release events of this nature, requires further development and possibly the implementation of alternative techniques well adapted for missing sources (e.g. Yu et al., 2021).


### 3.4.2 Feasibility of 1-day emission estimates - Upper Silesian Coal Basin, Poland (June, 2018)

The Upper Silesian Coal Basin (USCB) is one of the largest CH$_4$ emitting regions in Europe, with emissions originating from ~40 coal mines (EEA, 2021). The region extends from southern Poland across the border to Czechia where CH$_4$ is released from deep coal deposits and emitted to the atmosphere via ventilation shafts (Fiehn et al., 2020).


To evaluate the feasibility of the system to quantify regional CH$_4$ emission sources within a 24-hour window we performed a one-day inversion over the USCB. Results were compared with emission estimates derived using aircraft observations combined with Eulerian and Lagrangian dispersion models (Kostinek et al., 2021) and a mass balance approach (Fiehn et al., 2020). These studies used extensive flight data from the 6th June 2018 to derive regional CH$_4$ emission estimates of 0.42-0.48

Tg yr$^{-1}$. The CoMet v2 bottom-up inventory (Fiehn et al., 2020) was specifically compiled for the purpose of the flight campaign and estimated emissions in the region of 0.58 Tg yr$^{-1}$. Our results for the 6th of June 2018 estimated USCB emissions of 0.57 Tg yr$^{-1}$, compared to our prior estimate of 0.63 Tg yr$^{-1}$ (Figure 8). This shows good agreement with CoMet v2 and an improved agreement with the top-down estimates. From January-June 2019, posterior estimates (0.58±0.17 Tg yr$^{-1}$) remain low relative to the prior, however they increase in 2020 resulting in an average estimate for 2019-2020 of

0.62±0.19 Tg yr$^{-1}$ compared to a prior of 0.64±0.01 Tg yr$^{-1}$. This suggest that whilst emissions in the basin increased over the




simulation duration, they were consistently overestimated in the prior. The prior emissions do not consider daily variability, whilst considerable variability was estimated by the posterior ($1.7\pm0.5$ kt day$^{-1}$).

### 3.4.3 Detection limit of inversion system - Oil Fields, Algeria (2019-2020)

The CH$_4$ emissions from a point source release event from a well pad at the Hassi Messaoud oil field in Algeria (31.7°N,
5.9°E) from October 2019 until August 2020 were previously quantified (Varon et al., 2021). Using Sentinel-2 observations they derived mean emissions of $9.3\pm5.5$ t hr$^{-1}$. From our inversions, and using only dates where TROPOMI observations were available within 0.4° of where the leak occurred (21 days between 9$^{th}$ October, 2019 and 9$^{th}$ August, 2020), we found average CH$_4$ emissions within a 1°x1° domain of $24.1\pm3.5$ t hr$^{-1}$ (Figure 9b). After the leak was sealed average emissions decreased to $22.7\pm2.2$ t hr$^{-1}$. Assuming any difference in emissions between the two time periods was caused by the release
event, we estimate mean leak emissions of $1.4\pm0.9$ t hr$^{-1}$. This suggests some detection was made, but quantification was not accurate when compared to a previous study (Varon et al., 2021). It seems likely the magnitude of the leak ($<$10 t hr$^{-1}$) approaches the detection limit of the inversion performed here, and far exceeds the limit for accurate quantification. Additionally, the low number of observation days during the 10-month leak period (21 days), might have contributed to the lack of robust detection.


The Illizi Basin (28.3°N, 9.0°E) is one of the largest gas producing regions in Algeria and is currently undergoing planned expansions (Ouki et al., 2019). Results from a 3°x1.5° domain within the basin suggest average emissions are ~20% higher ($0.24\pm0.05$ Tg yr$^{-1}$) than those estimated by the prior inventory ($0.20\pm0.01$ Tg yr$^{-1}$) between 2019 and 2020 (Figure 9d). These results suggest the Illizi Basin is a larger source of CH$_4$ emissions than the Hassi Messaoud oil field ($0.21\pm0.03$ Tg yr$^{-1}$), although it should be noted the domain area is larger. As with the Hassi Messaoud oil field, with our system, it is not
possible to attribute the emission changes to a specific facility but rather to the entire region (~200 km$^2$).

### 3.4.4 Detection of unknown sources - Istanbul, Turkey (2020)

Istanbul is the most populous city in Europe, with prior CH$_4$ emission estimates of ~0.7 Tg yr$^{-1}$, making it one of the largest emitting regions of Europe. Prior information attributes 86% of those emissions to the solid waste and wastewater sector.
Inversion results from a 1°x1° domain centred around Istanbul (41.0°N, 29.0E°) showed an unexpected increase in emissions from July 2020 onwards, before which posterior ($0.68\pm0.10$ Tg yr$^{-1}$) emission estimates were in good agreement with the prior ($0.68\pm0.03$ Tg yr$^{-1}$) (Figure 10). From July to September 2020, these emissions increased by 42% to $0.97\pm0.30$ Tg yr$^{-1}$. The reason for this step change in emissions is unclear and, assuming the posterior estimates are robust, requires further investigation given the magnitude of the increase. Increased emissions are derived over a large area of the Istanbul domain;
however, given results from the Eagle Ford blowout region it is possible the estimated increase is from a point source. It is also unclear whether this is a new persistent emission source or if it only occurred over a period of several months.





**3.5 CH4 emissions during the COVID-19 period**

To evaluate the impact on anthropogenic $CH_4$ emissions from the global slowdown, caused by the COVID-19 pandemic, we compared posterior emissions from January to June of 2019 and 2020. Globally, average anthropogenic emissions for the 6-month period in 2020 ($359.5\pm22.0$ Tg yr$^{-1}$) are found to be 1.6% higher than for 2019 ($353.9\pm23.5$ Tg yr$^{-1}$) (Figure 11). These increased emissions contributed to the observed increased atmospheric growth rate between 2019 (10.0 ppb yr$^{-1}$) and 2020 (14.7 ppb yr$^{-1}$) (NOAA, 2021). Sector specific attribution shows the energy ($+2.7\pm1.6$ Tg yr$^{-1}$) and agriculture ($+2.0\pm0.5$ Tg yr$^{-1}$) sectors are the largest contributors to this increase, with smaller contributions from the waste ($+0.6\pm0.4$ Tg yr$^{-1}$) and other anthropogenic sources ($0.4\pm0.2$ Tg yr$^{-1}$).

When compared with 2019, anthropogenic $CH_4$ emissions in 2020 were larger pre-slowdown (January-February: $+5.6\pm0.0$ Tg yr$^{-1}$), considerably larger during the early stages of the slowdown (March-April: $+8.2\pm0.9$ Tg yr$^{-1}$) and only slightly larger in the latter months of the initial slowdown (May-June: $+3.2\pm0.0$ Tg yr$^{-1}$). This suggests, globally, the impact of the slowdown initially increased emissions and subsequently reduced them, although emissions for all 6 months were higher in 2020 than for 2019. This trend in emissions was mainly driven by energy sector emissions (January-February: $+2.4$ Tg yr$^{-1}$, March-April: $+4.7$ Tg yr$^{-1}$, May-June: $+0.9$ Tg yr$^{-1}$), whilst the agricultural sector showed a relatively consistent increase, relative to 2019, for all months.

When averaged over all 6 months, an increase in emissions between 2019 and 2020 was estimated in 6 out of 8 of the largest emitting regions, with the only exceptions being Pakistan ($-0.0$ Tg yr$^{-1}$) and Brazil ($-0.28$ Tg yr$^{-1}$). The largest increase was in China ($+2.6$ Tg yr$^{-1}$), of which, over half originated from the energy sector, specifically from the northern coal mining regions. The difference in emissions from China, relative to 2019, were the main driver for the global trend with increases pre-slowdown (January-February: $+3.9$ Tg yr$^{-1}$), large increases during the initial slowdown (March-April: $+6.0$ Tg yr$^{-1}$) and only small increases in the latter months (May-June: $+1.4$ Tg yr$^{-1}$). As with the global signal, this monthly variability is attributed to changes in energy sector emissions.

Emissions for 2020 from India were on average 0.8 Tg yr$^{-1}$ higher than for 2019, with noticeable large increases in emissions from the agricultural sector in June 2020 ($+1.3$ Tg yr$^{-1}$), which contributed to almost half of the global increase for June. The increased emissions in June mainly originated from the Uttar Pradesh region in north India. Similar increases in agricultural emissions are found over Bangladesh for June ($+1.3$ Tg yr$^{-1}$). Poor prior information in the region may have resulted in the misallocation of emissions which could have originated from the large Baghjan Oil Field blowout in Assam, India, in May/June 2020. Energy sector emissions from Indonesia were consistently higher in 2020 ($+0.2$ to $+0.6$ Tg yr$^{-1}$) and relatively unchanged for the remaining regions ($< \pm0.3$ Tg yr$^{-1}$).





Given the limitations of our system we have typically focused on anthropogenic emissions; however, natural fluxes were also derived. Global posterior results for the first half of 2020 show a reduction in both wetland (-0.4 Tg yr$^{-1}$) and fire (-1.8 Tg yr$^{-1}$) emissions when compared with 2019, with large monthly variability. The total global decrease in fire emissions is unchanged from the estimated prior emissions, taken from GFAS, which is based on satellite observations. The wetland emission change originates from South America, mainly from Brazil (-0.14 Tg yr$^{-1}$) and Argentina (-0.33 Tg yr$^{-1}$). These

reduced emissions were likely caused by large scale droughts which occurred in early 2020 (Marengo et al., 2021). Although the months simulated are not typically associated with the boreal northern hemisphere fire season, most of the reduction in biomass burning emissions came from Russia (-1.3 Tg yr$^{-1}$) and Canada (-0.53 Tg yr$^{-1}$). This change was caused by a particularly active arctic fire season in 2019 (Zhang et al., 2021) and large wildfires in northern Alberta in May 2019. Relative to 2019, increased fire emissions from Australia are derived for January 2020 (+2.6 Tg yr$^{-1}$). It is estimated that an

unusually intense bushfire season (Shiraishi and Hirata, 2021) resulted in the release of 330 kt $CH_4$ from Australia over the month of January alone, 220 kt $CH_4$ more than 2019. More specifically, the emissions were unusually large from New South Wales and Victoria.

A limitation of the current system is the use of a climatological OH sink, which is the primary oxidant for atmospheric $CH_4$.

Currently, OH is not included in the control vector and does not respond to changes in atmospheric chemistry. Formation pathways of OH are influenced by atmospheric $NO_x$ concentrations, which were estimated to have decreased during the slowdown period (Doumbia et al., 2021). Several simulations were performed using multiple chemistry schemes to assess the atmospheric impact of OH when using a slowdown adjusted emission scenario (Huijnen et al., 2021). Results show global OH decreases of 1-3% during the slowdown period, however a heterogenous spatial pattern is observed near the

surface with increased OH concentrations over some regions. This would suggest the 2020 increased emissions found here may be overestimated; however, the derived emission increases in January and February of 2020, relative to 2019, are unlikely to have been influenced by OH changes caused by the global slowdown. Future developments will include the addition of OH in the control vector and the use of an online OH loss rate derived using atmospheric chemistry, resulting in more accurate source/sink attribution.

**4 Conclusions**

We have investigated the feasibility to monitor $CH_4$ emissions using a global online high-resolution (~80km) short-window 4D-Var (24-hour) data assimilation system and satellite observations from multiple sensors. This system optimises both the initial 3D atmospheric concentration of $CH_4$ and surface fluxes, whilst implicitly accounting for transport errors associated with uncertainty in meteorological initial conditions. The prior emission errors were selected based on comparisons with

independent TCCON retrievals. We identify strengths and weaknesses of our inversion system by performing case study comparisons with other well-established flux estimation systems at a range of spatiotemporal scales.





Globally a small decrease in annual $CH_4$ emissions, relative to the prior, is estimated by the inversion for 2019 (~1%). At a national scale, we found decreased anthropogenic emissions from China (-5%) and India (-3%), with small increases from
USA (+0.5%) and Brazil (+1.3%) contributing to this change, this is in general agreement with a recent inverse study (Qu et al., 2021).

To evaluate the system at the regional and point scale, several anthropogenic case studies were selected (Table 1). Posterior estimates of anthropogenic sources with persistent emissions typically showed good agreement with previous studies. In
addition, the posterior quantification of emissions from a large biogenic source region, Lake Chad, compared well with a previous inversion study (Lunt et al., 2019).

We investigated the feasibility to quantify emissions at a high spatial, temporal and spatiotemporal resolution. Emissions from a well leak in the Hassi Messaoud oil field, which persisted for several months, were found to be at or around the
detection limit of the system (~9 $tCH_4$ $hr^{-1}$) and beyond the limit for accurate quantification. Similarly, emissions from a large well blowout in Eagle Ford were found to be misallocated to the surrounding region owing to poor prior information and too coarse model resolution. In contrast, inverse estimates from a known persistent point source, the Appin Mine, were found to be in good agreement with a previous top-down estimate (Varon et al., 2020). For a 1-day period over a large region, the Upper Silesian Basin, inverse estimates agreed well with previous studies, (Fiehn et al., 2020; Kostinek et al.,
2021). Overall, these case studies suggest our inverse system is suitable for regional scale (~100km²) emission quantification over a short time-period (24-hour), given sufficient satellite observations are available. Given adequate prior information our system is also capable of quantifying emissions from a persistent point source (e.g. Appin Mine, Australia).

Several previously undocumented $CH_4$ emission sources were also investigated, including an unknown release event from
the Bakken Formation. Prior emission estimates were persistently found to be underestimated by ~20% from the Illizi Basin between 2019 and 2020, possibly owing to an expansion in operations. Finally, a noticeable step change in emissions from Istanbul was observed from July 2020, when emissions increased by ~40%, the reason for which is unknown and would require further investigation.

The impact on $CH_4$ emissions from the global slowdown in response to COVID-19 was investigated using inversions from the first half of 2019 and 2020. The slowdown coincided with a year where $CH_4$ growth (14.7 ppb) was the largest since records began in the early 1980s. We found in the early part of 2020 atmospheric growth was, in part, driven by anthropogenic emissions which were larger than for 2019 (January to February: +5.6±0.0 Tg $yr^{-1}$). These emissions further increased during the early stages of the slowdown (March to April: +8.2±0.9 Tg $yr^{-1}$), almost half of which originated from
the energy sector in China. Had this trend continued the global growth rate for 2020 would have been even larger. However,

during the later months of the slowdown period emissions reduced, although were still slightly higher than 2019 values (May to June: $+3.2\pm0.4$ Tg yr$^{-1}$), suggesting the slowdown may have acted to reduce emissions, mainly from the energy sector. The overall impact of the global slowdown on $CH_4$ emissions is found to be small and the increased atmospheric growth is the result of a continued increasing trend in $CH_4$ emissions and possibly related to changes in atmospheric chemistry in response
to the slowdown (e.g. Stevenson et al., 2021). The reason for the observed variability in emissions is unclear, it is possible a reduction in energy demand resulted in increased venting of natural gas or a change in working practice led to an increase in fugitive emissions which subsequently fell below previous levels after several months of reduced demand.

Future developments will adapt the system for use to constrain $CO_2$ emissions based on a hybrid-ensemble system that will
extend the assimilation window and utilise observations of co-emitted species (e.g., $NO_2$, CO). Additionally, improved representation of biogenic fluxes as well as spatiotemporal correlations in the prior will provide more accurate posterior estimates and uncertainties. Finally, the current lack of error propagation across the 4D-Var windows, will be addressed in an upcoming version of the system and more dynamical approaches to automatically adjust inaccurate prior information will be implemented to better constrain missing and intermittent sources.

**Data Availability**

Data are available upon request to the corresponding authors.

**Author Contributions**

The methodology was developed by NB, JM, AA-P, GB, RE, ZK, AI. Model simulations were performed by JM. Validation against TCCON observations was performed by AA-P. Prior emissions and fire emission evaluation was prepared by MP,
AA-P. Analysis of the results was performed by JM, NB, AA-P, VH, AI. Preparation of TROPOMI observations was done by RR. All authors contributed to the preparation of the manuscript.

**Competing Interests**

The contact author has declared that neither they nor their co-authors have any competing interests.

**Acknowledgments**

The CoCO2 project has received funding from the European Union's Horizon 2020 Research and Innovation programme under Grant agreement No 958927.



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




**Table 1: Estimated prior and posterior emissions of CH₄ from several regions and events between 2018 and 2020. Comparison is made with existing case studies. Also given is the dominant source type (>50%) and the number of days when TROPOMI observations are made within 1° of the domain of interest. Values denoted by ± indicate standard deviation across all days.**

| Region | Lat | Lon | Dominant Source Type | Dates (where available) | No. TROPOMI observation on days (total: 485) | Prior Emissions (kt yr⁻¹) | Posterior Emissions (kt yr⁻¹) | Previous Estimates (kt yr⁻¹) |
|---|---|---|---|---|---|---|---|---|
| Permian Basin | 32.0°N | 103.0°W | Oil/Gas Field | Jan 2019-Sep 2020 | 237 | 1970±30 | 2290±470 | 2900 (Zhang et al., 2020) |
| Bakken Formation | 48.5°N | 103.0°W | Oil/Gas Field | Jan 2019-Sep 2020 | 93 | 1040±0 2019-only: 1040±0 | 1000±570 2019-only: 930±480 | 890 (Schneising et al., 2020) |
| Lake Chad | 14.3°N | 13.0°E | Agriculture/Wetlands | Jan 2019-Sep 2020 | 65 | 346±24 | 383±53 | No value given (Lunt et al., 2019) |
| Appin Colliery | 34.2°S | 150.8°E | Coal Mining | Jan 2019-Sep 2020 | 100 | 53±1 | 50±6 | 44-51 (Varon et al., 2020) |
| Eagle Ford | 28.9°N | 97.6°W | Blowout Event | Oct 2019-Nov 2019 | 15/45 (Oct/Nov 2019) | - | 22 (4°x4°: 892) | 242-534 (Cusworth et al., 2021) |
| Upper Silesian Coal Basin | 18.7°N | 50.0°E | Coal Mining | 6th June 2018 | 103 (total) | 627 | 572 | 423-581 (Fiehn et al., 2020; Kostinek et al., 2021) |
| Hassi Messaoud | 31.7°N | 5.9°E | Well Leak | Oct 2019-Aug 2020 | 21/306 | - | 28±29 | 81 (Varon et al., 2021) |
| Illizi Basin | | | Oil/Gas Field | Jan 2019-Sep 2020 | 172 | 203±5 | 236±47 | - |
| Istanbul | 41.2°N | 29.0°E | Waste | Jan 2019-Sept 2020 | 219 | Pre-July 2020: 681±34 Post-July 2020: 648±4 | Pre-July 2020: 677±103 Post-July 2020: 967±301 | - |


**Figure 1: a) Schematic of different resolutions used in the inversion shown by pseudo-data for 5 sectors. The magnitude of prior emissions at ~9 km (left) and those same emissions used as input to the forward model at ~25 km (middle). The inversion increment at ~80 km, resulting scaling factors are applied to all sectors within the grid cell, the boxes indicate relative contribution per sector (right). b) Schematic of inversion setup using the 24-hour window, correcting for the initial 3D state, emissions, and**
**initial conditions in the prior of the subsequent window.**




**Figure 2: a) Global annual mean prior CH₄ emissions for 2019 taken from CAMS. b) Difference between posterior and prior emissions averaged between January and June 2019, derived from the IFS inversion. c) Posterior adjustment, as a percentage of prior, in anthropogenic CH4 emissions per country.**

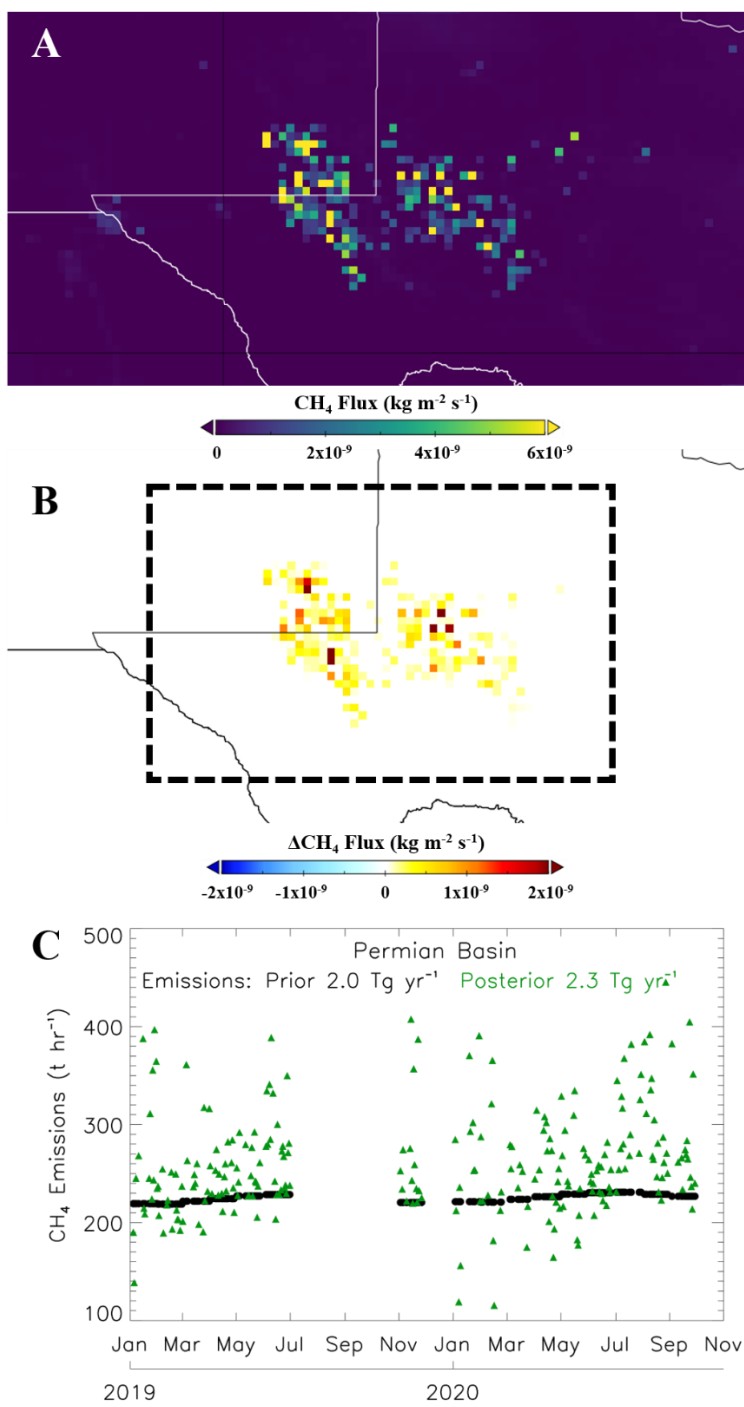


**Figure 3: a) Average prior Permian Basin CH₄ emissions for 2019. b) Average of posterior minus prior anthropogenic CH₄ emissions over the Permian Basin for January-June 2019, excluding days for which observations were not available. c) Time series of total prior (black circles) and posterior (green triangles) anthropogenic CH₄ emission estimates within the 6°x4° Permian Basin domain, centered around 32°N, 103°W (black box in b) for 2019-2020, excluding days for which observations were not available.**



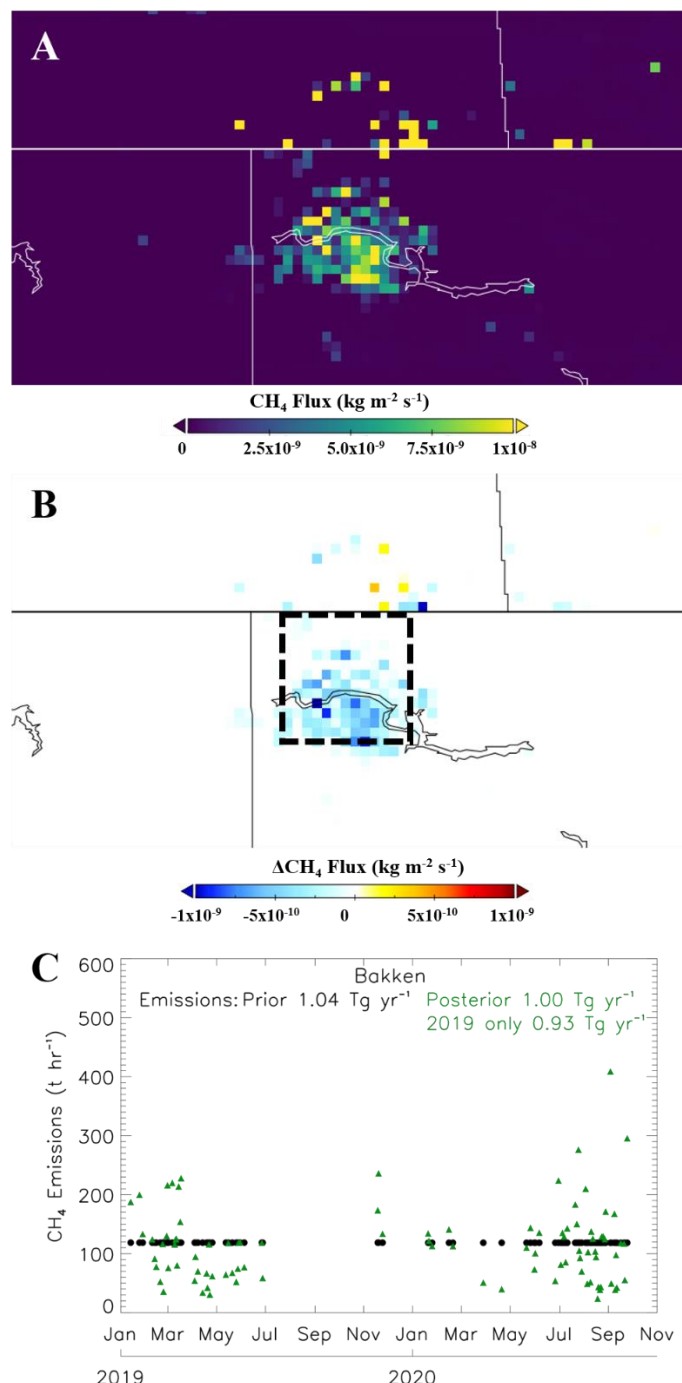


**Figure 4: a) Average prior Bakken CH4 emissions for 2019. b) Average of posterior minus prior anthropogenic CH₄ emissions over Bakken for January-June 2019, excluding days for which observations were not available. c) Time series of total prior (black circles) and posterior (green triangles) anthropogenic CH₄ emission estimates within the 1°x1° Bakken domain,  centred around 48.5°N, 103°W (black box in b) for 2019-2020, excluding days for which observations were not available.**





**Figure 5: a) The Lake Chad domain indicated by the black box (© Google Maps, 2021). b) Time series of total prior (black circles) and posterior (green triangles) CH₄ emission estimates within the 1°x1° domain, centred around 13.0°N, 14.3°E for 2019-2020, excluding days for which observations were not available. c) Average prior CH₄ emissions for 2019. d) Average posterior minus prior CH₄ emissions for January-June 2019, excluding days for which observations were not available.**



**Figure 6: a) The sector specific contribution to emissions within the Appin Colliery domain. b) Time series of total prior (black circles) and posterior (green triangles) CH₄ emission estimates within the domain for 2019-2020, excluding days for which observations were not available. c) Prior CH₄ emissions for January 2019, the white box denotes the grid cell used to estimate emissions. d) Average posterior minus prior CH₄ emissions for 2019, excluding days for which observations were not available.**





**Figure 7:** a) Prior (black circles) and Posterior (green triangles) anthropogenic $CH_4$ emission estimates, where observations are available, over an oil well blowout event in Eagle Ford, USA during October/November 2019 at the grid scale (a) and within a $4°x4°$ domain (b). The nearest date ($2^{nd}$ November) to the event, which occurred on the $1^{st}$ November, is indicated. Regional scaling factor values from the inversion for November $1^{st}$ (C), $2^{nd}$ (D) and $3^{rd}$ (E). Eagle Ford blowout site marked with an 'x' and $4°x4°$ domain denoted.





**Figure 8: a)** The Upper Silesian Coal Basin 1°x0.5° domain indicated by the white box, centred around 50.0°N, 18.7°E. Also shown are eleven major coal mines in the region (© Google Maps, 2021). **b)** Time series of total prior (black circles) and posterior (green triangles) CH₄ emission estimates within the domain for 2019-2020, where observations and inverse simulations were available. **c)** Prior total CH₄ emissions for 6th June 2018. **d)** Average posterior minus prior CH₄ emissions for 6th June 2018.






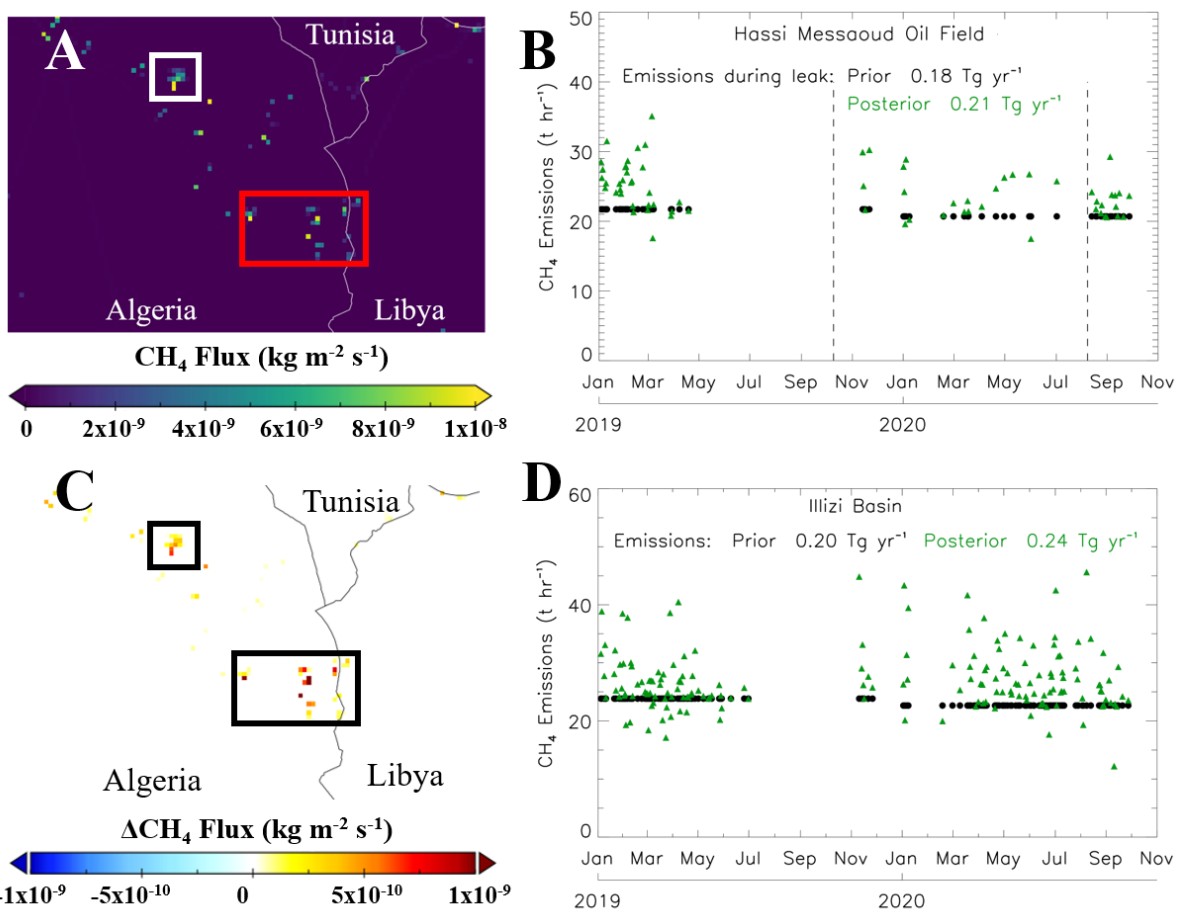

**Figure 9: a) Prior average anthropogenic CH₄ emissions for Eastern Algeria, 2019. The domains for the Hassi Massaoud oil field (1°x1°, white box) and part of the Illizi Basin (3°x1.5°, red box) are marked. Time series of total prior (black circles) and posterior (green triangles) CH₄ emission estimates within the Hassi Massaoud (b) and Illizi Basin (d) domains for 2019-2020, where observations and inverse simulations were available. c) Average posterior minus prior CH₄ emissions for 2019, using dates where nearby observations were available.**


**Figure 10: a) Prior CH₄ emissions within the Istanbul domain for September 2020. b) Time series of total prior (black circles) and posterior (blue triangles) CH₄ emission estimates within the 1°x1° domain, centred around 41°N, 29°E (white box in a) for 2019-2020, where observation and inverse simulations are available. Average posterior minus prior CH₄ emissions for May (c) and September (d) 2020, using dates where nearby observations were available.**





**Figure 11: Estimated national/regional average CH₄ emission change between 2020 and 2019 for January to June, derived using an IFS inversion for the largest emitters for a) Energy, b) Agriculture, c) Waste and d) Other Anthropogenic sources. e) Global change in sector specific monthly CH₄ emissions for the same period. f) National/regional change in total anthropogenic CH₄ emissions for the same period.**