# Peer review of "Quantification of methane emissions from hotspots and during COVID-19 using a global atmospheric inversion"

_Atmospheric Chemistry and Physics, 2021_

## Referee Comment (RC1)

Overview:
The manuscript "Quantification of methane emissions from hotspots and during COVID-19 using a global atmospheric inversion" by McNorton et al. describes the results of a high-resolution atmospheric inversion of methane (CH4) emissions during 2019 and 2020. There is focus of many individual case studies of various scales, and investigation into the effect of the COVID-19 pandemic on global and regional emissions of CH4.

Overall the manuscript is fairly well written, although there are some technical corrections necessary. The figures are clear and appropriate. The model simulations carried out for this work appear to produce useful and interesting results, and future improvements to the system will further refine such outputs. In the main text, details are often obscure and some sections need to be rewritten with more clarity. My main issue is that assertions are sometimes made without sufficient evidence to back them up, and I don't agree that the authors have clearly demonstrated that one of their main results is sufficiently robust.

If the revisions detailed below are sufficiently addressed, in particular those regarding the conclusions that the authors make regarding the effect of the global pandemic on methane emissions, I am happy for this manuscript to be published in ACP.

**Comments:**

Abstract and throughout: Many emission values are given as annual totals (e.g. 'CH4 emissions for 2020 were 5.7 Tg yr-1 (+1.6%) higher than for 2019) but I think that the differences in these totals must be based only on the first six months of each year, as the inversions do not cover the full years. This is misleading and should made clear throughout.

Line 16 - Without context the phrase 'basin-wide' is confusing. A gas basin? River basin? Wash basin?

Line 22 and later on: Your assertion that the the large atmospheric growth rate in 2020 would have occurred with or without the pandemic slowdown can not be supported for numerous reasons. The reasoning for this statement is not properly explained anywhere. As far as I can tell, it seems to be based on the fact that the global emission growth in May-June 2020 over 2019 is smaller than the growth in the pre-slowdown period in January-February 2020, and acts to cancel out the 'extra' emissions in March - April.

My issues with this logic are as follows. First, only the first half of the year 2020 has been modelled in this work, so no definitive conclusions about the whole year's growth rate can be made. Second, without carrying an inversion for a counterfactual world in which there was no pandemic (which is obviously not possible), you can't say what would have happened to emissions during summer 2020. It is possible that they would have been equal to, or lower than, those in 2019 and the global slowdown was in fact still acting to increase emissions during this time. You cannot therefore allocate any change in emission growth during this time to only the global slowdown. Third, much of China had lockdowns during January and February 2020, before the global slowdown began in earnest. Many of these were lifted in March and April. This Jan/Feb period therefore does not entirely represent 'business-as-usual' for comparison to later parts of the year. In my opinion these statements need to be more thoroughly examined and explained, or removed from the document.

Line 23: 'below expected pre-slowdown levels'. Again, this statement assumes that the observed emission growth in Jan/Feb is equivalent to an expected value for the rest of the year.

Line 24: 'small' in what sense? Emissions were higher in 2020 than in 2019 in each of these months. How are you quantifying the effect of the slowdown?

Line 25: Generally, descriptions of future work do not belong in an abstract.

Line 41: Is there any uncertainty included in the value of 14.7 ppb?

Line 43: Does this statement about venting/flaring conflict with the previous statement that oil and gas emissions reduced by 10% in 2020 (IEA)? It seems to, as written.

Line 46: I think that Weber et al. seem to suggest that the effect of changes to OH in 2020 on $CH_4$ have an upper bound of approximately 2 ppb on the observed growth rate. Since the difference in growth rates between 2020 and 2019 is approximately 4.7 ppb $yr^{-1}$, the OH effect is maybe not so small?

Line 47: The first part of this sentence is confusing. Do you mean that we have accurate measurements, or that theoretically, given accurate measurements, inverse modelling is possible? It should be rewritten.

Line 54: State the start date that SCIAMACHY data is available from, as you have for GOSAT.

Line 56 - 58: IASI measurements have been used in the inversion, and should therefore also be mentioned here.

Line 69: 'greater observability' - briefly explain why?

Introduction: The results of Forster et al. (2020) should be referenced somewhere.

Line 88: What is the justification for simulating a longer period during 2020 than in 2019? Is this taken into account when comparing e.g. the global annual total fluxes in the two years later on? (2019 posterior fluxes will have reverted to the prior for July - September whereas 2020 posterior fluxes will not have done so).

Line 117: Between each 24-hour window, the initial 3D mixing ratios are included in the state vector and therefore total mass of $CH_4$ is not conserved in the model. This is a justifiable consequence of the 4D-Var method with these short windows, but do you expect that it would affect your posterior flux estimations to a significant extent? If the system can 'reset' the mixing ratios to some extent every day, is it possible that some model-observation mismatch that are in reality due to emission changes can 'go missing' in the initial mixing ratios? How large were the prior uncertainties applied to the 3D grid and were error covariances included in this? This should be briefly discussed in the manuscript.

Line 149: It would be good to have a map of the applied observation uncertainties also included in the supplementary material if possible.

Line 149: Was the satellite data filtered in any way before use?

Line 150: Whilst I acknowledge that it might have been too much detail for this manuscript, it would generally be good to quantify the impact of the TROPOMI observations in the inversion over just using the IASI and GOSAT observations. Would the major conclusions about COVID-19, for example, have been any different without the TROPOMI data?

Line 170: How different were these values from those in the control simulation? Is the improvement from optimising the emissions significant relative to the observation uncertainty? Is the model performance degraded at any TCCON sites by optimising emissions?

Line 179 - 182: OK, I get that you want to only analyse properly-constrained grid cells. But does using this method have any impact on what you are quantifying? Are emission totals for 2019 and 2020 directly comparable, or do they have different spatial representations? Are regional and global totals in this manuscript comparable to other studies?

Line 184: Similarly - are posterior estimates for only the first six months of each year included here? Or do the two years have posterior totals included for different numbers of months? If only limited numbers of months are included for each year, what exactly does the value of 528.2 Tg $yr^{-1}$ in 2019 represent and is it really accurate to say that emissions in 2019 were 4.7 Tg $yr^{-1}$ smaller than in the prior? It is important to be clear with your language here.

Line 190: This figure suggests that the majority of countries' total anthropogenic emissions are quantified to within 1% by the prior emission inventory. Is this really likely, or is it a result of strict prior uncertainties applied to these countries in the inversion?

Line 192: What does 'other anthropogenic' cover? It should be noted somewhere, although not necessarily in this line.

Line 193: You state 'multiple inverse studies' and then only reference one. Add 'e.g.' or more references.

Line 195: Confused by reference to Outer Mongolia here for two reasons - first, there is not currently a state or country of this name. Do you mean the state of Mongolia? Second, in Figure 2b, there doesn't really seem to be any observable change to $CH_4$ fluxes in Mongolia.

Line 226: This trend doesn't sound that small to me (about 6% $yr^{-1}$)? Although it might not be significant given the data spread. I'm also not sure that a 300 kt change from the prior to the posterior should be described as considerable but a 150 kt/yr trend be described as small. Finally, I think units of the trend should be in kt $yr^{-2}$, or preferably Tg $yr^{-2}$.

Line 227: Different periods are covered in the two studies too. Is the sampled region the same between the studies?

Line 229: The reference from the introduction (Lyon et al., 2020) found reduced emissions from the Permian Basin in April & May 2020. Why not compare to this reference here too?

Line 233: I assume this means oil & gas production specifically in this region has increased?

Line 240: The phrasing here is very confusing: 'Our posterior results for 2019 … show a variable but positive trend'. Do you mean that the trend is variable? Or that the variability of the data is large but has a positive trend? Is the trend statistically significant? Generally, talking of trends when discussing a 12-month period is odd, and even if you're describing a trend between 2019 and 2020 this is still a very short period to derive trends over.

Line 242: The value found by Schneising et al. does have relatively large uncertainty attached, which your prior estimate and posterior estimates for 2019 and 2020 both fall well within.

Line 248: Please clarify somewhere which incident you mean? Incidents on September 4th in this file don't appear to be particularly large compared to those on other days, as far as I can tell.

Lines 251 - 262: I feel that this section concerning Lake Chad needs to be either expanded or removed, as it does not add much to the study in its current state. Comparing to a previous study that does not provide quantitative results for the region in question does not inform the reader of much. Perhaps contact the authors of that study? Meanwhile, the limited number of days with data assimilated does not provide information on the seasonality of local fluxes.

Line 340: and also noted that a greater number of days are included in deriving the emission rate for Illizi than Hassi Messaoud.

Line 351: Very low emission rates in early 2019 also notable?

Section 3.5: What do flux uncertainties in early paragraphs here represent?

Line 364: Again, this only holds if you assume that emission growth would have remained at the Jan/Feb value all year, which is not necessarily true. The slowdown might still have been increasing emissions in May & June relative to a world in which the pandemic did not happen.

Line 425: How do you know that it compared well?

Lines 452, 453 and 455: There is no evidence provided that the slowdown was the cause of reduced growth in emissions in May/June. Similarly, it can't be said that the overall impact of the slowdown was small as there is no counterfactual. Finally, if changes in the sink played a significant role, then it's even less possible to say with such certainty what impact the slowdowns had on methane emissions - perhaps emissions were in fact lower during March/April than in Jan/Feb but this could not be captured in the model.

Line 457: Has there been any research using bottom-up methods to compare to your results for emissions during the global slowdown? (E.G. the IEA data referenced in the introduction).

Line 466: It would be much more beneficial to the scientific community if data were put in a public repository.

Figure 2C: does the x-axis here show the prior or posterior annual emissions? And is it the actual annual emissions, or the first six months' emissions scaled to Tg $yr^{-1}$?

Figure 3 onwards: it might be beneficial to show prior uncertainty in these figures (with shading/dashed lines) if it does not affect clarity too much.

Figure 3 onwards: it would be helpful if the maps in these figures had an inset panel or similar, showing their location.

Figure 6A: 3D pie charts are a terrible way to display data, and the one here is certainly unnecessary. The 88.9% figure could just be stated in the main text, or a stacked bar chart could be used if you really want to plot this information.

**Technical corrections:**

Throughout: For some reason superscripts and subscripts have been omitted throughout the manuscript (e.g. those in $yr^{-1}$, $CH_4$, $CO_2$). Whilst not vital at this stage, the text would have been easier to read had they been included.

Throughout: the mixing of units through out the text again makes some of the discussion more difficult to follow. You should not be switching so often

between mass units of t and kt along with Tg, particularly as this is often within the same sentence or figure panel.

Throughout: links to section titles etc. in the main text should be capitalised (e.g. Section 3.1.1, Supplementary Figure 1, Table 1).

Line 14: newly available -> newly-available

Line 16: form -> from

Line 30: CO2 -> carbon dioxide ($CO_2$)

Line 39: remove 'when'

Line 40: oil producing -> oil-producing

Line 64: remove 'a first version of the'

Line 64: ECMWF should be defined during first use.

Line 69: fix 'allows to benefit'

Line 91: 3-hourly

Line 137: delete 'a' before ~80 km.

Line 139: should this read something like 'limited occurrences of co-located emissions from…'?

Line 175: Should be 'business-as-usual' as this is adjectival.

Line 181: 'To this aim' -> 'To this end'/'With this aim in mind', etc.

Line 201: Previously-documented

Line 202: Barre et al. (2020)

Line 223: Change to 'The uncertainty value here represents the standard deviation of the daily fluxes and not the posterior uncertainty'.

Line 267: remove unnecessary comma after 'and'.

Line 271: remove unnecessary commas after 'derived' and 'fugitive-only'.

Line 272: business-as-usual

Line 359: Add + before 0.4.

Line 575: Double spaced.

Figure 6 caption and others: sector-specific

**REFERENCE:**

Forster, P.M., Forster, H.I., Evans, M.J. *et al.* Current and future global climate impacts resulting from COVID-19. *Nat. Clim. Chang.* **10,** 913–919 (2020). https://doi.org/10.1038/s41558-020-0883-0

---

## Referee Comment (RC2)

*Quantification of methane emissions from hotspots and during COVID-19 using a global atmospheric inversion*

**General comments**

The paper deals with the estimates of CH4 surface emissions by using a short window (24-h) 4D-Var global inverse modelling system based on the ECMWF Integrated Forecasting System (IFS) within the Bayesian framework. The system uses solely satellite retrievals of the total column of CH4 concentrations (XCH4) to constrain the surface fluxes of CH4. First, the authors performed a suite of sensitivity experiments to identify an appropriate prior flux uncertainty. Thus, by using several prior estimates, they carried out inversions and used their forward model to compute the relevant optimised concentrations that are confronted to XCH4 measurements from the global TCCON network. Second, by using their best prior flux information, they performed estimates of CH4 fluxes at global and regional scales for some months of the years 2019 and 2020. Furthermore, they investigated the feasibility of their system in addressing different CH4 emission hotspots in several parts of the world. Finally, the authors assessed the impact of COVID-19 lockdown on the fate of CH4 surface emissions.

Results show that the system is able to estimate the fluxes at both regional scales provided appropriate prior fluxes and relative dense observations. Their investigation of the feasibility of the system in addressing the CH4 emission is more challenging. However, results show that the system is capable of detecting part of the blowout events when enough observations are available. There, maybe the limit of the system does not depend only on the prior information, as the authors often stated in the text, but also on the lack of pertinent observations during such rapid events due to the smoothed nature of the satellite retrievals (in time and space). In fact, for such rapid events, the use of in situ surface measurements (if available and especially at a high temporal resolution) in addition to the satellite retrievals may give better estimates.

They are two points to clarify:

1) To compensate for the lack of uncertainties in the inverted fluxes, the authors considered only pixels having a reasonable number of satellite retrievals to calculate the posterior fluxes. Doing so is reasonable, but this can introduce some uncertainties when comparing your results to other published estimates.

2) Why did the authors perform the inversions only for some months of the two years? Indeed, doing this is enough for the COVID-19 lockdown, but this may also introduce some uncertainties when comparing the results of this study at annual scale to the previous analyses.

This is a comprehensive work on the estimates of CH4 surface emissions at global, regional scales and source point events. I have very much appreciated the part of the work that investigates the feasibility of the system in addressing source point emissions. Indeed, this is a challenging subject, but the results seem to indicate that the system is promising. I have also appreciated the fact that the authors underline the strengths and weaknesses of their system (e.g., missing uncertainty on the posterior fluxes) by giving some ideas for its improvement. As an example, they plan to increase the size of the short assimilation window of their system. Here, it is not easy to know if the increase of this window would be beneficial? However it is worth implementing it. As I mentioned above, maybe also to envisage the inclusion of in situ surface measurements in the system that may enrich the

satellite retrievals in terms of peaks (high temporal resolution in situ data) for some of these rapid source point events. Moreover, in general, the inclusion of in situ measurements may help to correct some biases in the satellite retrievals.

Overall, the methodology is sound and quite well described and especially the results are well presented and discussed. Moreover, the manuscript is well written and easy to read. After the clarifications of both the two issues highlighted above and some suggestions in my specific comments below, I would recommend the manuscript for publication in ACP

**Specific comments**

**Abstract**

Lines 17-18: …, *but without accurate prior uncertainty information, were not well quantified.'* Accurate? This is a bit misleading. This would mean that you have compared this uncertainty to a reference value. Maybe use as in the text: ' .. with appropriate prior uncertainty or your best prior information

Lines 18-20: As already mentioned in my general comments, I do not understand why you performed the inversions only part of these years (i.e., 2019 and 2020). The comparison of the posterior estimates to the prior ones for the same period is fine, but the numbers for each of these years may be uncertain owing to the missing months.

**1. Introduction**

Lines 26-27: '*Changes in atmospheric chemistry, not investigated here, may have contributed to the observed growth in 2020'*: How? The decrease of OH? Please elaborate.

Line 80: '*…. at a high spatial and temporal resolution'* by at both high spatial and temporal resolutions ?

**2. Methods**

Lines 88-90: *'These were performed from January to June of 2019 and January to September of 2020..'.* I have not understood why the other months of these two years are not considered. Please elaborate

Line 133: Why do you consider the flux of the model LPJ-WHyme (page 96) and then you use the uncertainty from WetCHARTs. It is possible to consider both the mean estimate of CH4 emissions from WetCHARTs and the uncertainty as you do here

Lines 149-151: '*TROPOMI uncertainties provided as part of the CH4 product were applied within the minimisation routine and averaging kernels were use*d.' Give a reference of this data. May be the reference like Otto Hasekamp et al., (2017) [https://sentinel.esa.int/documents/247904/2476257/Sentinel-5P-TROPOMI-ATBD-Methane-retrieval] or an updated version?

Line 150: *'Additional CH4 observations from the …'* by Additional XCH4 observations from the …

**3. Results**

Lines 158: '… most of countries...' by most of the countries in the world.. ?

Line 170: '*When evaluating XCH4 concentrations simulated with optimised emissions, the all-site average lowest standard error (6.8 ppb), absolute mean bias (7.52 ppb) and highest R-value (0.74) was found for the mapped prior error described in section 2.2.2.*'. Please give the range of R-value instead of only the highest R-value, otherwise give the lowest R-value.

**3.2. Global estimates.**

In this paragraph, I have not understood why you performed the inversions only for some months of the years 2019 and 2020. You compared your numbers to those of other studies in literature. Your numbers might include some additional uncertainties due the lack of these months. Moreover, the way you sample your optimised fluxes can introduce additional uncertainty in the numbers. In fact, what can be the contribution of the pixels discarded to the total numbers? I do not question the way you sampled your data, just be aware that doing so can introduce additional uncertainty and this needs to be quantified/mentioned in the paper. In fact, a rough estimation of the uncertainty due to your sampling method can be estimated by considering all the pixels to generate the numbers and compare them to those derived from your sampling method. I do not want to repeat these above remarks in other parts of the paper, hence please consider them and address them in the other parts of the paper when relevant.

**3.3. Emission estimates for Regions and Point Sources**

Line 205: '*To filter posterior estimates which provided little or no updated information we omitted daily grid cells associated with poor observation constraints (see supplement figure 1).*' Again, how much these pixels may contribute to the numbers? This remark is valid for the other parts of the study.

Lines 226-227: '*While it is difficult to diagnose the cause of the difference in posterior estimates, one possibility is the larger prior uncertainty used in Zhang et al. (2020)*'. Yes, but not only the prior information can explain this. In fact, the inverse modelling system, as it is built, with a good observational coverage it may be possible to infer the fluxes from the surface whatever the prior information. Hence, here both prior information and observations (here about 50% of the data are used and maybe no information in some pivotal areas) are meaningful to explain the deficiencies.

**3.3.4 Point source emissions- ….** Bracket after Australia to be deleted

**3.5. CH4 emissions during the COVID-19 period**

Page 390: '*These 390 reduced emissions were likely caused by large scale droughts which occurred in early 2020 (Marengo et al., 2021*'. The droughts events may have decreased the water table?

**4.    Conclusion**

Last paragraph: lines 459-464

Please adapt the text to the CH4 inverse modelling system. Here, you may mention only the new developments of the CO2 inverse modelling system that are relevant for the CH4 inversion system

**Tables**

OK

**Figures**

Figure 2:  C). The title of y-axis should be '% Change in Emissions  100.*(Posterior-Prior)/Prior

**References**

Overall fine, only at page 16 move Courtier et al. 1994 after Cheewaphongphan et al. 2019

**Supplement**

Figure 1: Quality flags. Please specify (>50% or larger ?)

---

## Community Comment (CC1)

**REVIEW**

*title*: Quantification of methane emissions from hotspots and during COVID-19 using a global atmospheric inversion
*author*: McNorton et al.
*DOI:* https://doi.org/10.5194/acp-2021-1056

*OVERVIEW*

With this paper the researchers want to validate a method to quantity methane emissions. They used inverse modelling of in-situ and TROPOMI satellite observations to quantify methane hotspot emissions, to subsequently compare them with results of existing case studies. A forward model is used to translate previous methane emissions into atmospheric methane concentrations. Then they applied a 4D-Var Inverse model to detect methane emission hotspots based on methane concentration forecasts from IFS.

Overall, the paper is very comprehensive and the research is conducted well. In particular, the forward model seems to work correctly as there is no (large) overestimation of emissions compared to previous results. According to Cheewaphongphan et al. (2019), overestimation of inverse modelling (top-down) methods can be largely explained by errors in emission estimations of bottom-up approaches, like the forward model in this study. The comparison of their own results with the existing case studies is very elaborate and documented in a structured way. The separate page of figures for every part of the results makes it convenient to interpret and to compare between different case studies.

Nonetheless, I have my doubts about the novelty of this research. There are many examples of other papers using the same method to model methane emissions, see my major arguments below. On the bright side, what is very interesting about this paper specifically is that it compares emissions for a regular year with a year of COVID-19 pandemic slowdown. The researchers expected that the effect of the global slowdown can be compared with mitigation strategies to decrease greenhouse gas emissions, but during the COVID-19 slowdown period methane emissions continued to rise.

In my opinion, a sufficient explanation of this unexpected result is missing. Without this explanation, it looks like the aim of the research – which they described in the introduction – has not been achieved. Throughout the paper, it is more common for assertions to be made without sufficient or unclear explanations. Therefore, some added analysis and detailed explanations are needed.

If the proposed changes and additional explanations are adequately incorporated into a revisited version, I would recommend this paper to be published in the journal of Atmospheric Chemistry and Physics.

*MAJOR ARGUMENTS*

*1.* First of all, in this paper a 24-hour data assimilation window is used in the Inverse method. Another paper about implementation of four-dimensional data assimilation stated that "4D-Var using a 6- or 12-hour window performed better than 3D-Var over a 2-week assimilation period, whereas 4D-Var using a 24-hour window did not" (Rabier et al., 2000). This means using a 24-hour window is no improvement over already existing methods, so there should be a strong reason why the paper under review isn't able to use this shorter window. Barré et al. (2021) also performed 4D-Var, but here they did use a 12-hour window. From the paper under review it is not clear why they did not use this shorter window.

Using a larger 4D assimilation window means that data is combined over a larger time span. Apparently, a 24-hour window is too large and thus not specific for one moment in time anymore. By averaging over a too long period, the time component is eliminated which makes the method move more towards a 3D method. But for a good functioning 3D method, this 24-hour data assimilation period is in turn way to short.

Consequently, all results of the paper could be affected by the choice of window length, since more averaging possibly means less accurate detection of methane emissions peaks. The researchers must have a strong reason why to use a 24-hour assimilation window, because it is unlikely to yield the most reliable results. Therefore, I argue that it is necessary to perform all the model simulations again using a shorter assimilation window, unless the researchers can provide a very solid explanation on why that is not possible.

*2.* My second issue is that I doubt about the novelty of this work. The paper of Barré et al. (2021) claims to be novel with a method that is highly similar to the method from the paper under review: it also used observations collected by TROPOMI and IFS methane forecasts produced by CAMS. The use of 4D-Var systems is not new, since Rabier et al. (2000) already described operational implementation of 4D-Var assimilation. A quick search yielded more papers that used a 4D-Var System to model methane emissions (e.g. Meirink et al., 2008; Van Peet et al., 2021; Yu et al., 2021).

However, new about this paper is the comparison of methane emissions between the COVID-19 and pre-COVID-19 situation. Most of the paper, though, is about testing the performance of their method and not the aforementioned comparison. If this comparison is supposed to be the aspect that distinguishes this paper from others, details and explanations are lacking in the conclusion (see also my next argument).

To overcome this issue of novelty I provide two options:

(1) If the novelty of the paper is about differences between the method of the paper under review and the method of other papers using 4D-Var systems, this has to be proved by explicitly pointing out these differences (at least with papers: Barré et al., 2021; Meirink et al., 2008; Van Peet et al., 2021; Yu et al., 2021). Now, it is not clear that there are obvious differences in methodology.

(2) If the novelty of the paper is about the comparison between the COVID-19 and pre-COVID-19 situation, the conclusion section needs to be rewritten. The presented results are not enough supported by argumentation. The results show methane emissions in 2020 to be higher than expected, but a solid explanation is missing. There has to be either a physical explanation or limitations of the method plays a role.

Concluding: one of these two scenarios has to be the case and need to be fixed as explained above. If not, then I strongly doubt about the novelty of this paper.

3.        Thirdly, I am concerned about the lack of attention paid to the atmospheric sink of methane. The paper of Maasakkers et al. (2019) used a method that I think can be seen as the precursor of the method under review, but in addition it did include an OH depending changing sink for atmospheric methane. A changing sink means that for every assimilation step new atmospheric OH concentrations are used. The amount of methane oxidation is determined by the NOx – OH – CH4 reactions, which means that the available amount of OH for methane oxidation depends on NOx emissions (Stevenson et al., 2021). I think this atmospheric chemistry is very interesting during the COVID-19 period, because reduced NOx emissions induced by the global slowdown thus probably lead to an increase in methane concentrations. In contrast, the paper under review used a constant climatological OH sink and thus the strength of the atmospheric methane sink is not depending on atmospheric chemistry.

Unexpected are the results that suggest that the methane emissions are higher in the COVID-19 situation than the period before. In the reasoning of the paper, they referred to two different sources. Stevenson et al. (2021) shows that the results of these higher than expected methane emissions can be explained by the use of a changing OH depending sink in the atmosphere, because this over time decreasing sink is strong enough to explain the increase in methane emissions. The paper of Weber et al. (2020) suggest that the effect of this OH sink is too small to explain the excess in methane emissions.

The paper under review does not decide which of these two contrasting result is the most likely. Independ of the possible effect of an OH based changing atmospheric sink, the researchers did not manage to fully explain the higher than expected methane emissions. Because, according to the aim of the research, this is the most important conclusion, I think there is a need to share their opinion about the strength of an OH depending atmospheric sink. If needed, they have to perform additional analysis to quantify the effect of a changing sink. If they doubt this effect to be large enough to explain the higher than expected methane emissions, they definitely have to come up with other possible reasons. Without a plausible explanation, the main result of the research is not sufficiently supported.

*MINOR ARGUMENTS*

*1.*        The reduced anthropogenic activities because of the COVID-19 pandemic, gave the researchers the possibility to look into the effects of potential climate mitigation strategies to decrease greenhouse gas emissions (Diffenbaugh et al., 2020). The paper and the source referred to does not explain why the researchers believe this to be a legitimate analogy. It seems a reasonable comparison, but in the conclusion it appeared not to be true: the methane emissions increased during the COVID-19 period, whereas the purpose of climate mitigation strategies is to reduce these emissions. In the paper, no further reflection is made on this comparison. I recommend explanation for this, as it is an interesting question why this analogy not seems to work.

*2.* In the method section is stated that "prior emissions errors are assumed to be independent between 24-hour inversion cycles". This assumption is said to be made because not enough is known about temporal error correlations. If this assumption is not valid and errors are dependent, there may be biases in the results, especially when the period coved by the model is extended. This extension of the modelled period is exactly wat is done with the comparison between the pre COVID-19 and during COVID-19 period. It would be good to indicate how bad this assumption is expected to affect the results by applying a range of error correlations and compare these results to the original results.

*3.* In the method section is stated that "posterior errors in methane emissions and 3D state are not propagated forward across data assimilation cycles". This shortcoming is said to be a technical limitation of the system and will be addressed in subsequent versions. This limitation can cause a bias in the results, but the researchers did not indicate how large they expect this effect to be. I recommend to perform some additional model runs with distorted initial posterior emissions and 3D states, to determine the effect on the final results compared to the original results.

*MINOR ISSUES*

*line 38* "The change in energy and fuel demand is estimated to have reduced oil and gas CH4 emissions by 10 % for 2020 when compared to 2019 (IEA, 2021)" The reference is a figure that does not contain information about this estimated 10% reduction. Pleas correct reference.

*line 61 / line 148 / line 202* Reference to Barré et al. is wrong, year of publication is 2021 instead of 2020.

*line 69* "For this paper, the focus on CH4 emissions allows to benefit from greater observability from remote-sensing (compared to CO2)" Why is this? Please explain.

*line 121* "background errors for the meteorological variables at initial time are constructed based on a climatology, and therefore are not flow-dependent" Please Explain, unclear what you mean.

*Line 126* "… at the relatively high increment resolution used (~80km) CH4 sectors are rarely collocated." Is this true? I think in a quare of 80 times 80 kilometres often multiple sectors occur.

*line 132* "Globally, constant wetland uncertainties were estimated at 58%, taken as the standard deviation from the WetCHARTs ensemble (Bloom et al., 2017)." From this paper it is not evident where the researches get this 58% from. Please explain.

*line 140* "Total grid cell uncertainties, used in the control vector, were calculated with the error propagation method." Is this a common method? Explanation or reference is lacking.

*line 160* "…, we perform simulations from January, when slowdown restrictions were limited to China, to June for 2019 and 2020." Period January to June in 2019 is clear why, but which period in 2020 exactly and why?

*line 171* "All subsequent experiments used the mapped prior uncertainty, typically ranging from 50-150%." What is the mapped prior uncertainty? Please explain.

*line 192* "we distributed total posterior emissions into 6 sector specific categories; energy, agriculture, waste, other anthropogenic, wetlands and fires". What is included in sector 'other anthropogenic'? Please explain.

*line 385*      "Given the limitations of our system we have typically focused on anthropogenic emissions…". How is the system limited? Why can that be fixed by focussing on only anthropogenic emissions? Please explain.

*line 459*      "Future developments will adapt the system for use to constrain CO2 emissions based on a hybrid-ensemble system that will extend the assimilation window and utilise observations of co-emitted species" What is a hybrid-ensemble system? Why is extension of the assimilation window a good thing? Please explain.

*figure 1*      Part B would be very useful to explain in a bit more detail how the 4D-Var system works, but in the text there is no reference to this figure. I like to see some explanation in the text about this figure.

*figures 2-10*      It would be good to make the link between the text and the figures a bit more clear when reading the paper, for example make it very obvious that there is one page with figures for every part of the results. In addition, the section names/numbers can be included in the description of the figures.

*figures 2-10*      *It is not clear why f*or some case studies a pi-chart and overview map is provided and for others not. An overview of the setting of the case study area and a graph of the relative contributions of all the sector emissions would probably be good to have for all the case studies.

*SOURCES  (NOT REFERENCED IN PAPER)*

Meirink, J. F., Bergamaschi, P., and Krol, M. C. (2008). Four-dimensional variational data assimilation for inverse modelling of atmospheric methane emissions: method and comparison with synthesis inversion, *Atmos. Chem. Phys.*, 8, 6341–6353, https://doi.org/10.5194/acp-8-6341-2008.

van Peet, J., Houweling, S., Marshall, J., Nunez Ramirez, T., and Segers, A. (2021). Inverse modelling of global methane emissions using TROPOMI, *EGU General Assembly 2021*, online, 19–30 Apr 2021, EGU21-14510, https://doi.org/10.5194/egusphere-egu21-14510.

Yu, X., Millet, D. B., and Henze, D. K. (2021). How well can inverse analyses of high-resolution satellite data resolve heterogeneous methane fluxes? Observing system simulation experiments with the GEOS-Chem adjoint model (v35), *Geosci. Model Dev.*, 14, 7775–7793, https://doi.org/10.5194/gmd-14-7775-2021.

---

## Author Comment (AC1)

"This review was prepared as part of graduate program course work at Wageningen University, and has been produced under supervision of Prof Wouter Peters. The review has been posted because of its good quality, and likely usefulness to the authors and editor. This review was not solicited by the journal."

The authors are grateful for the comments made on the manuscript and have discussed with Wouter Peters the concerns raised by the contributor. Wouter has agreed the major suggested changes are not required. Below, in brief, we address these concerns and their relevance to this manuscript. Whilst several misunderstandings are addressed below several comments are constructive and will be included in a revised version of the manuscript with the response to the official reviews.

REVIEW

title: Quantification of methane emissions from hotspots and during COVID-19 using a global atmospheric inversion
author: McNorton et al.
DOI: https://doi.org/10.5194/acp-2021-1056

OVERVIEW

With this paper the researchers want to validate a method to quantity methane emissions. They used inverse modelling of in-situ and TROPOMI satellite observations to quantify methane hotspot emissions, to subsequently compare them with results of existing case studies. A forward model is used to translate previous methane emissions into atmospheric methane concentrations. Then they applied a 4D-Var Inverse model to detect methane emission hotspots based on methane concentration forecasts from IFS.

Overall, the paper is very comprehensive and the research is conducted well. In particular, the forward model seems to work correctly as there is no (large) overestimation of emissions compared to previous results. According to Cheewaphongphan et al. (2019), overestimation of inverse modelling (top-down) methods can be largely explained by errors in emission estimations of bottom-up approaches, like the forward model in this study. The comparison of their own results with the existing case studies is very elaborate and documented in a structured way. The separate page of figures for every part of the results makes it convenient to interpret and to compare between different case studies.

Nonetheless, I have my doubts about the novelty of this research. There are many examples of other papers using the same method to model methane emissions, see my major arguments below. On the bright side, what is very interesting about this paper specifically is that it compares emissions for a regular year with a year of COVID-19 pandemic slowdown. The researchers expected that the effect of the global slowdown can be compared with mitigation strategies to decrease greenhouse gas emissions, but during the COVID-19 slowdown period methane emissions continued to rise.

In my opinion, a sufficient explanation of this unexpected result is missing. Without this explanation, it looks like the aim of the research – which they described in the introduction – has not been achieved. Throughout the paper, it is more common for assertions to be made without sufficient or unclear explanations. Therefore, some added analysis and detailed explanations are needed.

If the proposed changes and additional explanations are adequately incorporated into a revisited version, l would recommend this paper to be published in the journal of Atmospheric Chemistry and Physics.

As mentioned above, Wouter Peters has confirmed to the authors that the major changes suggested are either not relevant to improving the manuscript or are beyond the scope of the work. Therefore, whilst we have updated the manuscript based on several minor comments the major comments will not be addressed for the reasons given below.

MAJOR ARGUMENTS

1. First of all, in this paper a 24-hour data assimilation window is used in the Inverse method. Another paper about implementation of four-dimensional data assimilation stated that "4D-Var using a 6- or 12-hour window performed better than 3D-Var over a 2-week assimilation period, whereas 4D-Var using a 24-hour window did not" (Rabier et al., 2000). This means using a 24-hour window is no improvement over already existing methods, so there should be a strong reason why the paper under review isn't able to use this shorter window. Barré et al. (2021) also performed 4D-Var, but here they did use a 12-hour window. From the paper under review it is not clear why they did not use this shorter window.

Using a larger 4D assimilation window means that data is combined over a larger time span. Apparently, a 24-hour window is too large and thus not specific for one moment in time anymore. By averaging over a too long period, the time component is eliminated which makes the method move more towards a 3D method. But for a good functioning 3D method, this 24-hour data assimilation period is in turn way to short.

Consequently, all results of the paper could be affected by the choice of window length, since more averaging possibly means less accurate detection of methane emissions peaks. The researchers must have a strong reason why to use a 24-hour assimilation window, because it is unlikely to yield the most reliable results. Therefore, I argue that it is necessary to perform all the model simulations again using a shorter assimilation window, unless the researchers can provide a very solid explanation on why that is not possible.

The manuscript of Rabier et al. (2000) focuses on 4D-Var for meteorological applications and not for greenhouse gases. The non-linear nature of meteorological processes demands a short-window length, whereas greenhouse gas inversions are usually performed using offline transport and using much longer window-lengths (monthly to yearly). In this study the window-length is relatively short compared to most global GHG inversions. A limitation of the study is in fact the window length being relatively short and not being too long. As discussed in the manuscript future developments aim to extend this window length without interfering with the short-window required for meteorology. As shown in the results, the short-window length limits the number of days where inversions can be performed for specific case studies resulting from sparse observations. A further reduction in the window-length would decrease the observational constraints further and therefore be unsuitable.

2. My second issue is that I doubt about the novelty of this work. The paper of Barré et al. (2021) claims to be novel with a method that is highly similar to the method from the paper under review: it also used observations collected by TROPOMI and IFS methane forecasts produced by CAMS. The use of 4D-Var systems is not new, since Rabier et al. (2000) already described operational implementation of 4D-Var assimilation. A quick search yielded more papers that used a 4D-Var System to model methane emissions (e.g. Meirink et al., 2008; Van Peet et al., 2021; Yu et al., 2021).

However, new about this paper is the comparison of methane emissions between the COVID-19 and pre-COVID-19 situation. Most of the paper, though, is about testing the performance of their method and not the aforementioned comparison. If this comparison is supposed to be the aspect that distinguishes this paper from others, details and explanations are lacking in the conclusion (see also my next argument).

To overcome this issue of novelty I provide two options:

(1) If the novelty of the paper is about differences between the method of the paper under review and the method of other papers using 4D-Var systems, this has to be proved by explicitly pointing out these differences (at least with papers: Barré et al., 2021; Meirink et al., 2008; Van Peet et al., 2021; Yu et al., 2021). Now, it is not clear that there are obvious differences in methodology.

(2) If the novelty of the paper is about the comparison between the COVID-19 and pre-COVID-19 situation, the conclusion section needs to be rewritten. The presented results are not enough supported by argumentation. The results show methane emissions in 2020 to be higher than expected, but a solid explanation is missing. There has to be either a physical explanation or limitations of the method plays a role.

Concluding: one of these two scenarios has to be the case and need to be fixed as explained above. If not, then I strongly doubt about the novelty of this paper.

4D-Var inversions are indeed well studied and the methodology available in the literature is extensive. Our manuscript does not claim to be the first study to perform 4D-Var inversions for $CH_4$. However, we believe it to be the first study to perform inversions at high-resolution on a global scale using online transport and TROPOMI data. In particular, an online transport configuration allows better representation of the transport error since uncertainties in the initial meteorological fields are accounted for. Further novelty comes from the cases and time-period of study as mentioned by the contributor. In reference to Barré et al. (2021), it is a very different study in that they perform forecast experiments and look at observation-model departure statistics to detect potentially anomalous emissions. Their study is not a 4D-Var inversion and the only major similarity is the use of the IFS system. In the manuscript the evolution of 4D-Var inversions of $CH_4$ is discussed and several previous studies are mentioned which have informed the developments presented.

3. Thirdly, I am concerned about the lack of attention paid to the atmospheric sink of methane. The paper of Maasakkers et al. (2019) used a method that I think can be seen as the precursor of the method under review, but in addition it did include an OH depending changing sink for atmospheric methane. A changing sink means that for every assimilation step new atmospheric OH concentrations are used. The amount of methane oxidation is determined by the NOx – OH – CH4 reactions, which means that the available amount of OH for methane oxidation depends on NOx emissions (Stevenson et al., 2021). I think this atmospheric chemistry is very interesting during the COVID-19 period, because reduced NOx emissions induced by the global slowdown thus probably lead to an increase in methane concentrations. In contrast, the paper under review used a constant climatological OH sink and thus the strength of the atmospheric methane sink is not depending on atmospheric chemistry.

Unexpected are the results that suggest that the methane emissions are higher in the COVID-19 situation than the period before. In the reasoning of the paper, they referred to two different sources. Stevenson et al. (2021) shows that the results of these higher than expected methane emissions can be explained by the use of a changing OH depending sink in the atmosphere, because this over time decreasing sink is strong enough to explain the increase in methane emissions. The paper of Weber et al. (2020) suggest that the effect of this OH sink is too small to explain the excess in methane emissions.

The paper under review does not decide which of these two contrasting result is the most likely. Independ of the possible effect of an OH based changing atmospheric sink, the researchers did not manage to fully explain the higher than expected methane emissions. Because, according to the aim of the research, this is the most important conclusion, I think there is a need to share their opinion about the strength of an OH depending atmospheric sink. If needed, they have to perform additional analysis to quantify the effect of a changing sink. If they doubt this effect to be large enough to explain the higher than expected methane emissions, they definitely have to come up with other possible reasons. Without a plausible explanation, the main result of the research is not sufficiently supported.

We believe the concern raised here is partly valid, although it is already addressed and the challenges of including OH are discussed in the manuscript. Including varying OH either requires online chemistry which would increase the computational cost of the system significantly, making it no longer viable to run for extensive periods. Alternatively, an offline OH field could be used for 2019 and 2020 based on full chemistry forward simulations; however, the uncertainty of the derived OH fields is considerable (see contrasting results from Stevenson et al. 2021 and Weber et al. 2020). As discussed in the manuscript we believe the influence of OH variability during the pandemic may have influence $CH_4$ lifetime and we include reference to a recent study using the IFS to investigate these changes. Including, and even remarking on, such changes is beyond the scope of this study; however, it will be investigated in the future. Several explanations for the trend in emissions are indeed mentioned in the manuscript, however remarking on human activities in detail would be unwise given the focus of the study is to investigate where and when emissions change and less so on socio-economic drivers of the changes.

MINOR ARGUMENTS

1. The reduced anthropogenic activities because of the COVID-19 pandemic, gave the researchers the possibility to look into the effects of potential climate mitigation strategies to decrease greenhouse gas emissions (Diffenbaugh et al., 2020). The paper and the source referred to
does not explain why the researchers believe this to be a legitimate analogy. It seems a reasonable
comparison, but in the conclusion it appeared not to be true: the methane emissions increased during the COVID-19 period, whereas the purpose of climate mitigation strategies is to reduce these
emissions. In the paper, no further reflection is made on this comparison. I recommend explanation
for this, as it is an interesting question why this analogy not seems to work.

As discussed above the socio-economic intricacies of the influence of COVID-19 are not the aim of this manuscript. The relation of COVID-19 to climate mitigation strategies focuses on a step change in emissions, these could either be a decrease or increase, either way it serves as a good testbed to explore shifts in emissions.

2. In the method section is stated that "prior emissions errors are assumed to be independent between 24-hour inversion cycles". This assumption is said to be made because not enough is known
about temporal error correlations. If this assumption is not valid and errors are dependent, there may be biases in the results, especially when the period coved by the model is extended. This extension of the modelled period is exactly wat is done with the comparison between the pre COVID-
19 and during COVID-19 period. It would be good to indicate how bad this assumption is expected to
affect the results by applying a range of error correlations and compare these results to the original
results.

With no available estimate of these correlations, various assumptions would be speculative. This also ties into the point below.

3. In the method section is stated that "posterior errors in methane emissions and 3D state are not propagated forward across data assimilation cycles". This shortcoming is said to be a technical
limitation of the system and will be addressed in subsequent versions. This limitation can cause a
bias in the results, but the researchers did not indicate how large they expect this effect to be. I recommend to perform some additional model runs with distorted initial posterior emissions and 3D
states, to determine the effect on the final results compared to the original results.

Such analysis of error correlations is not possible with the system as described owing to limitations in the uncertainty calculations, this is mentioned throughout the manuscript. This is an area of future development. It is unclear what is meant by "distorted initial posterior emissions and 3D states". If this is perturbing the prior with information from the posterior of the previous window then as discussed throughout the manuscript this requires substantial modifications to the system. As discussed with Wouter Peters, further simulations should not be conducted as it is beyond the scope of this study.

MINOR ISSUES

Several of the suggested minor issue changes have been uptaken and will be included in the revised manuscript.

line 38 "The change in energy and fuel demand is estimated to have reduced oil and gas CH4 emissions by 10 % for 2020 when compared to 2019 (IEA, 2021)" The reference is a figure that does
not contain information about this estimated 10% reduction. Pleas correct reference.

line 61 / line 148 / line 202 Reference to Barré et al. is wrong, year of publication is 2021 instead
of 2020.

line 69 "For this paper, the focus on CH4 emissions allows to benefit from greater

observability from remote-sensing (compared to CO2)" Why is this? Please explain.

line 121 "background errors for the meteorological variables at initial time are constructed based on a climatology, and therefore are not flow-dependent" Please Explain, unclear what you
mean.

Line 126 "… at the relatively high increment resolution used (~80km) CH4 sectors are rarely collocated." Is this true? I think in a quare of 80 times 80 kilometres often multiple sectors occur.

line 132 "Globally, constant wetland uncertainties were estimated at 58%, taken as the standard deviation from the WetCHARTs ensemble (Bloom et al., 2017)." From this paper it is not
evident where the researches get this 58% from. Please explain.

line 140 "Total grid cell uncertainties, used in the control vector, were calculated with the error propagation method." Is this a common method? Explanation or reference is lacking.

line 160 "…, we perform simulations from January, when slowdown restrictions were limited to China, to June for 2019 and 2020." Period January to June in 2019 is clear why, but which period in
2020 exactly and why?

line 171 "All subsequent experiments used the mapped prior uncertainty, typically ranging from 50-150%." What is the mapped prior uncertainty? Please explain.

line 192 "we distributed total posterior emissions into 6 sector specific categories; energy, agriculture, waste, other anthropogenic, wetlands and fires". What is included in sector 'other anthropogenic'? Please explain.

line 385 "Given the limitations of our system we have typically focused on anthropogenic emissions…". How is the system limited? Why can that be fixed by focussing on only anthropogenic
emissions? Please explain.

line 459 "Future developments will adapt the system for use to constrain CO2 emissions based on a hybrid-ensemble system that will extend the assimilation window and utilise observations
of co-emitted species" What is a hybrid-ensemble system? Why is extension of the assimilation window a good thing? Please explain.

figure 1 Part B would be very useful to explain in a bit more detail how the 4D-Var system works, but in the text there is no reference to this figure. I like to see some explanation in the text
about this figure.

figures 2-10 It would be good to make the link between the text and the figures a bit more clear when reading the paper, for example make it very obvious that there is one page with figures for
every part of the results. In addition, the section names/numbers can be included in the description
of the figures.

figures 2-10 It is not clear why for some case studies a pi-chart and overview map is provided and
for others not. An overview of the setting of the case study area and a graph of the relative contributions of all the sector emissions would probably be good to have for all the case studies.

SOURCES (NOT REFERENCED IN PAPER)

Meirink, J. F., Bergamaschi, P., and Krol, M. C. (2008). Four-dimensional variational data assimilation
for inverse modelling of atmospheric methane emissions: method and comparison with synthesis
inversion, Atmos. Chem. Phys., 8, 6341–6353, https://doi.org/10.5194/acp-8-6341-2008.

van Peet, J., Houweling, S., Marshall, J., Nunez Ramirez, T., and Segers, A. (2021). Inverse modelling
of global methane emissions using TROPOMI, EGU General Assembly 2021, online, 19–30 Apr 2021,
EGU21-14510, https://doi.org/10.5194/egusphere-egu21-14510.

Yu, X., Millet, D. B., and Henze, D. K. (2021). How well can inverse analyses of high-resolution satellite
data resolve heterogeneous methane fluxes? Observing system simulation experiments with the GEOS-Chem adjoint model (v35), Geosci. Model Dev., 14, 7775–7793, https://doi.org/10.5194/gmd-
14-7775-2021.

---

## Author Response (AR1)

**Response to reviewers - Quantification of methane emissions from hotspots and during COVID-19 using a global atmospheric inversion**

The authors would like to thank both reviewers for taking the time to go into the details of the manuscript and believe the comments provide constructive criticisms which have improved the quality of the manuscript. Please find below a point-by-point response to each review comment to accompany the updated and track change manuscript.

Review 1

Overview:

The manuscript "Quantification of methane emissions from hotspots and during COVID-19 using a global atmospheric inversion" by McNorton et al. describes the results of a high-resolution atmospheric inversion of methane ($CH_4$) emissions during 2019 and 2020. There is focus of many individual case studies of various scales, and investigation into the effect of the COVID-19 pandemic on global and regional emissions of $CH_4$.

Overall the manuscript is fairly well written, although there are some technical corrections necessary. The figures are clear and appropriate. The model simulations carried out for this work appear to produce useful and interesting results, and future improvements to the system will further refine such outputs. In the main text, details are often obscure and some sections need to be rewritten with more clarity. My main issue is that assertions are sometimes made without sufficient evidence to back them up, and I don't agree that the authors have clearly demonstrated that one of their main results is sufficiently robust.

If the revisions detailed below are sufficiently addressed, in particular those regarding the conclusions that the authors make regarding the effect of the global pandemic on methane emissions, I am happy for this manuscript to be published in ACP.

We thank the reviewer for their comments and agree with the issues raised. We have attempted to improve the clarity of some sections. We have attempted to include additional evidence or remove previous claims where evidence was lacking. The details are described below.

Abstract and throughout: Many emission values are given as annual totals (e.g. '$CH_4$ emissions for 2020 were 5.7 Tg yr-1 (+1.6%) higher than for 2019) but I think that the differences in these totals must be based only on the first six months of each year, as the inversions do not cover the full years. This is misleading and should made clear throughout.

We agree and whilst it is confusing to present emissions as 6mo$^{-1}$ we agree that the results are misleading by providing values as annual totals. We have updated the text where appropriate stressing that these are emissions for the first half of the year for both 2020 and 2019. We hope this adds more clarity and transparency to the results.

Line 16 - Without context the phrase 'basin-wide' is confusing. A gas basin? River basin? Wash basin?

Reference to basin has been removed, now it just refers to regional and point specific.

Line 22 and later on: Your assertion that the large atmospheric growth rate in 2020 would have occurred with or without the pandemic slowdown can not be supported for numerous reasons. The reasoning for this statement is not properly explained anywhere. As far as I can tell, it

seems to be based on the fact that the global emission growth in May-June 2020 over 2019 is smaller than the growth in the pre-slowdown period in January-February 2020, and acts to cancel out the 'extra' emissions in March - April.  My issues with this logic are as follows. First, only the first half of the year 2020 has been modelled in this work, so no definitive conclusions about the whole year's growth rate can be made. Second, without carrying an inversion for a counterfactual world in which there was no pandemic (which is obviously not possible), you can't say what would have happened to emissions during summer 2020. It is possible that they would have been equal to, or lower than, those in 2019 and the global slowdown was in fact still acting to increase emissions during this time. You cannot therefore allocate any change in emission growth during this time to only the global slowdown. Third, much of China had lockdowns during January and February 2020, before the global slowdown began in earnest. Many of these were lifted in March and April. This Jan/Feb period therefore does not entirely represent 'business-as-usual' for comparison to later parts of the year. In my opinion these statements need to be more thoroughly examined and explained, or removed from the document.

We agree the conclusions reached may have been lacking in explanation. The main motivation for comments on the impact of the slowdown come from a relative comparison of January/February for both 2020 and 2019 alongside a comparison with the remaining 4 months. From this the results indicate larger 2020 emissions than 2019 both pre-slowdown and during. Given the relative change for both time periods is similar we conclude the slowdown impact is small, however as mentioned we may be neglecting other influences and have added this caveat to the text (2nd point) and as discussed below, for China, February might not be a suitable 'business-as-usual' month.

In response to the 1st point, as previously mentioned we have updated the text to explain better the results only represent half a year and not a full annual growth nor do they extend beyond 12 months into the continued 2021 slowdown.

In response to the 3rd point. According to the COVID-19 stringency index, cited within the manuscript, the slowdown period in China begun on 23rd January, so we agree that taking February emissions as business as usual may not be appropriate for China. However, for the rest of the world measures were typically introduced in March. The relative differences in derived emissions for January 2019 and 2020 when compared with February are very similar, further suggesting that the impact of the slowdown was small (assuming other factors did not contribute). The caveat to our approach is then, assuming other factors beyond the slowdown did not have significant impact on emissions, we conclude the continuation of emission increases were not significantly impacted by the slowdown either nationally for China or globally. We have included this in the text reiterating the point of the reviewer that direct comparisons are challenging and attributing changes to the slowdown alone neglects other factors.

 Line 23: 'below expected pre-slowdown levels'. Again, this statement assumes that the observed emission growth in Jan/Feb is equivalent to an expected value for the rest of the year.

The assumption is not that Jan/Feb is equivalent to the expected value for the rest of the year, but rather the relative change from 2019 to 2020 for each month is expected to be the same or similar in a business-as-usual scenario, which is what we find. We have added several references in the text to the assumptions made in drawing this conclusion.

Line 24: 'small' in what sense? Emissions were higher in 2020 than in 2019 in each of these months. How are you quantifying the effect of the slowdown?

Whilst emissions were found to be higher in 2020 than 2019 the relative change in the pre-slowdown months (January and February (exc. China)) is similar to the change during slowdown months, hence why we conclude the impact is small. This is the case both globally and for most major emitting regions (see figure 11). We agree this is not well described in the text and have added this into the main manuscript.

Line 25: Generally, descriptions of future work do not belong in an abstract.

Thanks, this has been removed.

Line 41: Is there any uncertainty included in the value of 14.7 ppb?

We have now included the reported uncertainty.

Line 43: Does this statement about venting/flaring conflict with the previous statement that oil and gas emissions reduced by 10% in 2020 (IEA)? It seems to, as written.

It does conflict with the other findings and we have included this now with "An alternative hypothesis is that reduced demand could have…". This is a hypothesis given here and not supported by other studies.

Line 46: I think that Weber et al. seem to suggest that the effect of changes to OH in 2020 on CH4 have an upper bound of approximately 2 ppb on the observed growth rate. Since the difference in growth rates between 2020 and 2019 is approximately 4.7 ppb yr-1, the OH effect is maybe not so small?

We agree and have updated the text to state Weber et al. show changes in OH alone could not account for the observed growth, although the importance of the OH trend is discussed in detail later in the manuscript and we recommend it is explored further.

Line 47: The first part of this sentence is confusing. Do you mean that we have accurate measurements, or that theoretically, given accurate measurements, inverse modelling is possible? It should be rewritten.

We agree this is confusing and have rewritten to explain that assuming we have accurate measurements then inverse modelling is possible.

Line 54: State the start date that SCIAMACHY data is available from, as you have for GOSAT.

Now added.

Line 56 - 58: IASI measurements have been used in the inversion, and should therefore also be mentioned here.

Now added.

Line 69: 'greater observability' - briefly explain why?

This has been slightly rephrased to better explain what is meant.

Introduction: The results of Forster et al. (2020) should be referenced somewhere.

We agree and this has now been added.

Line 88: What is the justification for simulating a longer period during 2020 than in 2019? Is this taken into account when comparing e.g. the global annual total fluxes in the two years later on? (2019 posterior fluxes will have reverted to the prior for July - September whereas 2020 posterior fluxes will not have done so).

For global comparisons between 2020 and 2019 we only use this first 6 months as described in the relevant sections. The additional months for 2020 were only used for the case studies.

Line 117: Between each 24-hour window, the initial 3D mixing ratios are included in the state vector and therefore total mass of CH4 is not conserved in the model. This is a justifiable consequence of the 4D-Var method with these short windows, but do you expect that it would affect your posterior flux estimations to a significant extent? If the system can 'reset' the mixing ratios to some extent every day, is it possible that some model-observation mismatch that are in reality due to emission changes can 'go missing' in the initial mixing ratios? How large were the prior uncertainties applied to the 3D grid and were error covariances included in this? This should be briefly discussed in the manuscript.

We agree this is a limitation and the total mass is not conserved as a result. For observations made nearby sources we would expect this impact to be small; however, on a larger spatial scale the impact could be considerable. Further investigation into the impact will be done in future work which will consider a long-window 4D-Var (e.g., several weeks or months) that improves mass conservation. The prior uncertainties in the 3D-state were temporally fixed but spatially varying based on an ensemble of forecasts, which is used in the CAMS operational system. A comment on this has been added to the text. We have also included a new figure to the supplementary material showing the prior uncertainty in the 3D-state.

Line 149: It would be good to have a map of the applied observation uncertainties also included in the supplementary material if possible.

We agree that quantification of these observation uncertainties was lacking and have instead added a value to the text.

Line 149: Was the satellite data filtered in any way before use?

Yes, the satellite quality flag is used in the assimilation as described in the Massart paper referenced. We have included a comment on quality flags.

Line 150: Whilst I acknowledge that it might have been too much detail for this manuscript, it would generally be good to quantify the impact of the TROPOMI observations in the inversion over just using the IASI and GOSAT observations. Would the major conclusions about COVID-19, for example, have been any different without the TROPOMI data?

We agree this is an interesting area to explore and was recently discussed by Qu et al. 2021 (Global distribution of methane emissions: a comparative inverse analysis of observations from the TROPOMI and GOSAT satellite instruments). In their study, despite TROPOMI having ~100 times more observations, it is found to have less impact than GOSAT. However, their system operated at a much coarser resolution so a direct comparison may not be appropriate. The question of how much TROPOMI data informed our results is indeed interesting, followup studies will investigate this further by removing sensors from inversions to test the sensitivity.

Line 170: How different were these values from those in the control simulation? Is the improvement from optimising the emissions significant relative to the observation uncertainty? Is the model performance degraded at any TCCON sites by optimising emissions?

A subsection of these comparisons are shown in the supplementary material, although not stated in the text the standard error in the control is 7.1 ppb, relative to 6.7 ppb in the inversion. The significance of this is difficult to quantify; however, given systematic TCCON uncertainty most simulations fall within this uncertainty range. The model performance is degraded when using the mapped prior for 7/21 sites; however there is always at least one prior error which outperforms the control. We therefore conclude that whilst the inversion improves the comparison, the prescribed prior error must be reasonable.

Line 179 - 182: OK, I get that you want to only analyse properly-constrained grid cells. But does using this method have any impact on what you are quantifying? Are emission totals for 2019 and 2020 directly comparable, or do they have different spatial representations? Are regional and global totals in this manuscript comparable to other studies?

For case studies we include "observable" days for transparency. We find that, as expected, days without observations have posterior values close to the prior. For comparison of 2019 and 2020 the spatial coverage of observations is typically similar for both years. Within a one-month period, in which the posterior estimates are averaged, the entire globe is well observed. Assuming there is not large variability within each month, the two years should be comparable. The regional and global totals are comparable to other studies and typically only show a small but important deviation from the bottom-up CAMS inventory. We have stated similarities with other studies e.g. Deng et al., 2021.

Line 184: Similarly - are posterior estimates for only the first six months of each year included here? Or do the two years have posterior totals included for different numbers of months? If only limited numbers of months are included for each year, what exactly does the value of 528.2 Tg yr-1 in 2019 represent and is it really accurate to say that emissions in 2019 were 4.7 Tg yr-1 smaller than in the prior? It is important to be clear with your language here.

We agree, our original statement was confusing and, as mentioned above, these comparisons are for just the first 6 months of each year and we have improved our description in the text.

Line 190: This figure suggests that the majority of countries' total anthropogenic emissions are quantified to within 1% by the prior emission inventory. Is this really likely, or is it a result of strict prior uncertainties applied to these countries in the inversion?

For the regions shown (the 8 major emitting regions), indeed 4 do fall within 1% of the prior estimate when averaged over a 6-month period. It is feasible that the total prior over this period is reasonably accurate, although, perhaps more likely the prior uncertainty at a daily timescale should indeed be inflated beyond the values used (as discussed for several of the case studies). Without sufficient prior information this is indeed a problem for inversion studies and there is on-going work in the COCO-2 project to further develop these prior uncertainties.

Line 192: What does 'other anthropogenic' cover? It should be noted somewhere, although not necessarily in this line.

We agree and have now added this into the text.

Line 193: You state 'multiple inverse studies' and then only reference one. Add 'e.g.' or more references.

Updated

Line 195: Confused by reference to Outer Mongolia here for two reasons - first, there is not currently a state or country of this name. Do you mean the state of Mongolia? Second, in Figure 2b, there doesn't really seem to be any observable change to CH4 fluxes in Mongolia

This is incorrectly referenced in the text and should refer to the province of Inner Mongolia, the text has been updated.

Line 226: This trend doesn't sound that small to me (about 6% yr-1)? Although it might not be significant given the data spread. I'm also not sure that a 300 kt change from the prior to the posterior should be described as considerable but a 150 kt/yr trend be described as small. Finally, I think units of the trend should be in kt yr-2, or preferably Tg yr-2.

We agree that this is not such a small trend and the statistical significance should be remarked on. We have now added this. The units are also updated.

Line 227: Different periods are covered in the two studies too. Is the sampled region the same between the studies?

The sampled region is the same, we have intentionally selected the same domain.

Line 229: The reference from the introduction (Lyon et al., 2020) found reduced emissions from the Permian Basin in April & May 2020. Why not compare to this reference here too?

Additional comparison has now been included with the Lyon et al. study. With some similarities and differences highlighted.

Line 233: I assume this means oil & gas production specifically in this region has increased?

Yes, this is specific to this region.

Line 240: The phrasing here is very confusing: 'Our posterior results for 2019 ... show a variable but positive trend'. Do you mean that the trend is variable? Or that the variability of the data is large but has a positive trend? Is the trend statistically significant? Generally, talking of trends when discussing a 12-month period is odd, and even if you're describing a trend between 2019 and 2020 this is still a very short period to derive trends over.

We agree this is confusing and have re-worded it to explain that the data is variable but has positive growth. This is essentially a trend over a 1.5 year period Jan 2019-Jun 2020, although we understand this is unclear in the way it is currently worded. We believe the analysis of a trend period is dependent on the contributing factors and here, given the rapid expansion of activities it is reasonable to infer a trend. We have modified the text for a clearer explanation.

Line 242: The value found by Schneising et al. does have relatively large uncertainty attached, which your prior estimate and posterior estimates for 2019 and 2020 both fall well within.

Like our study, Schneising et al. range in values are based on the variability between days/observations and not just the uncertainty in the observations/methodology. We do agree

that these lead to a large range of values, but it is difficult to deduce what fraction of this is flux uncertainty and what is caused by emission variability.

Line 248: Please clarify somewhere which incident you mean? Incidents on September 4th in this file don't appear to be particularly large compared to those on other days, as far as I can tell.

This is incident ID 48996, Break, Leak or Malfunction of Equipment. They quote 5,000m$^3$ of gas released, making it one of the larger incidents in the report. Our estimates are, however, noticeably larger than those reported. We didn't want to attribute this event with the estimated emission increases but merely consider it a possibility.

Lines 251 - 262: I feel that this section concerning Lake Chad needs to be either expanded or removed, as it does not add much to the study in its current state. Comparing to a previous study that does not provide quantitative results for the region in question does not inform the reader of much. Perhaps contact the authors of that study? Meanwhile, the limited number of days with data assimilated does not provide information on the seasonality of local fluxes.

We agree this section is lacking quantative comparisons with the other study. However, given the period is different a direct comparison of values might not be sensible, instead the emphasis is on the prior inventories used generally being low. The intention of this section is to show the possibility to quantify natural emissions even given the caveats; however, despite this being a heavily caveated result we still wanted to include it to show the possibilities of the system.

Line 340: and also noted that a greater number of days are included in deriving the emission rate for Illizi than Hassi Messaoud.

This is mainly caused by the Illizi being a larger domain, the observable days which are available should still be enough to draw the conclusion that the Ilizi Basin is a larger CH$_4$ source.

Line 351: Very low emission rates in early 2019 also notable?

We agree this is notable, the reason for this is unknown and it would be more difficult to draw conclusions on missing sources than the larger increase seen in 2020. Also, the 2020 increase seems to be more sustained.

Section 3.5: What do flux uncertainties in early paragraphs here represent?

L223 states the uncertainties represent the daily variability not the posterior uncertainty. We understand this is unclear and wanted to add it at the start of the analysis for clarity. We have updated the text to highlight this.

Line 364: Again, this only holds if you assume that emission growth would have remained at the Jan/Feb value all year, which is not necessarily true. The slowdown might still have been increasing emissions in May & June relative to a world in which the pandemic did not happen.

We agree this message is slightly confusing and have reword several sections. Importantly, the comparison is made with 2019 and not with January and February of 2020. Each month is compared with the equivalent month of the previous year to remove the seasonality. The hypothesis is a relative change for a given month in 2019 and 2020 should be similar across all months unless the slowdown acted to alter emissions. Given the difference between 2019 and

2020 is similar for each individual month, we therefore conclude the impact is small. There are several assumptions made here though and we have tried to clarify this in the text.

Line 425: How do you know that it compared well?

The bias alone in the prior is consistent, so whilst not fully quantitative the prior (which is similar) is still adjusted in the same direction.

Lines 452, 453 and 455: There is no evidence provided that the slowdown was the cause of reduced growth in emissions in May/June. Similarly, it can't be said that the overall impact of the slowdown was small as there is no counterfactual. Finally, if changes in the sink played a significant role, then it's even less possible to say with such certainty what impact the slowdowns had on methane emissions - perhaps emissions were in fact lower during March/April than in Jan/Feb but this could not be captured in the model.

We agree that it is not conclusive that the further increase in emissions derived in Mar/Apr relative to the increase seen in Jan/Feb is attributed to the slowdown, however, this does seem the most plausible explanation. The same is true of the relative decrease in the longer-term increase seen for May/Jun. We believe the slowdown is the most likely explanation for this; however, as this is not conclusive, we have included this uncertainty in the updated manuscript. Assuming no other change, there is no clear shift in emissions over the entire 4-month period (Mar-Jun), we believe this is a reasonable assumption excluding the possible influence of OH changes.

Line 457: Has there been any research using bottom-up methods to compare to your results for emissions during the global slowdown? (E.G. the IEA data referenced in the introduction).

The IEA report a 10% reduction in emissions, however this is not seen here. To our knowledge there are no reliable mapped bottom-up estimates, but studies using simple assumptions about proxy data, e.g. Forster et al. (2020) estimate a slight reduction in emissions during the early stages of the slowdown. These have been included in the text. Doumbia et al. 2021 also provide an updated emission dataset; however, this does not include CH4.

Line 466: It would be much more beneficial to the scientific community if data were put in a public repository.

We agree; however, given the large size of the model files it is not feasible to store them in a public repository, all results are hosted on ECMWF MARS archive and available to those with access, we are happy to retrieve specific data from the archive for the wider-scientific community who cannot access MARS.

Figure 2C: does the x-axis here show the prior or posterior annual emissions? And is it the actual annual emissions, or the first six months' emissions scaled to Tg yr-1?

We agree this is unclear and have updated the caption, it is prior emissions and the emissions are based on the first 6 months alone, the flux is now given in units of months.

Figure 3 onwards: it might be beneficial to show prior uncertainty in these figures (with shading/dashed lines) if it does not affect clarity too much.

We agree it would be helpful; we have updated figures 3-10 with prior uncertainties

Figure 3 onwards: it would be helpful if the maps in these figures had an inset panel or similar, showing their location.

We agree and for several of these plots we have now added a zoomed out map of the region inset to the plots.

Figure 6A: 3D pie charts are a terrible way to display data, and the one here is certainly unnecessary. The 88.9% figure could just be stated in the main text, or a stacked bar chart could be used if you really want to plot this information.

Whilst pie charts can be an effective way to visualise the components of a whole we have acted on the reviewers opinion and removed the pie chart and replaced it with a regional map.

All technical corrections were applied to the manuscript and once again we thank the reviewer for their detailed feedback.

**Review 2**

The paper deals with the estimates of CH4 surface emissions by using a short window (24-h) 4D-Var global inverse modelling system based on the ECMWF Integrated Forecasting System (IFS) within the Bayesian framework. The system uses solely satellite retrievals of the total column of CH4 concentrations (XCH4) to constrain the surface fluxes of CH4. First, the authors performed a suite of sensitivity experiments to identify an appropriate prior flux uncertainty. Thus, by using several prior estimates, they carried out inversions and used their forward model to compute the relevant optimised concentrations that are confronted to XCH4 measurements from the global TCCON network. Second, by using their best prior flux information, they performed estimates of CH4 fluxes at global and regional scales for some months of the years 2019 and 2020. Furthermore, they investigated the feasibility of their system in addressing different CH4 emission hotspots in several parts of the world. Finally, the authors assessed the impact of COVID-19 lockdown on the fate of CH4 surface emissions.

Results show that the system is able to estimate the fluxes at both regional scales provided appropriate prior fluxes and relative dense observations. Their investigation of the feasibility of the system in addressing the CH4 emission is more challenging. However, results show that the system is capable of detecting part of the blowout events when enough observations are available. There, maybe the limit of the system does not depend only on the prior information, as the authors often stated in the text, but also on the lack of pertinent observations during such rapid events due to the smoothed nature of the satellite retrievals (in time and space). In fact, for such rapid events, the use of in situ surface measurements (if available and especially at a high temporal resolution) in addition to the satellite retrievals may give better estimates.

They are two points to clarify:

1) To compensate for the lack of uncertainties in the inverted fluxes, the authors considered only pixels having a reasonable number of satellite retrievals to calculate the posterior fluxes. Doing so is reasonable, but this can introduce some uncertainties when comparing your results to other published estimates.

This is a good point, and specifically true for studies which use observations from different sources as many of those used in the text do. We have included this additional source of uncertainty to comparisons in the text.

2) Why did the authors perform the inversions only for some months of the two years? Indeed, doing this is enough for the COVID-19 lockdown, but this may also introduce some uncertainties when comparing the results of this study at annual scale to the previous analyses.

We agree and this was also mentioned by the other reviewer. We only ran for a limited number of months due to the computational cost of the system. We agree providing the results as an annual emission as a result is misleading and have updated this in the text. Furthermore, as mentioned it makes comparisons with other studies slightly less consistent and this also now noted in the text.

This is a comprehensive work on the estimates of CH4 surface emissions at global, regional scales and source point events. I have very much appreciated the part of the work that investigates the feasibility of the system in addressing source point emissions. Indeed, this is a challenging subject, but the results seem to indicate that the system is promising. I have also appreciated the fact that the authors underline the strengths and weaknesses of their system (e.g., missing uncertainty on the posterior fluxes) by giving some ideas for its improvement. As an example, they plan to increase the size of the short assimilation window of their system. Here, it is not easy to know if the increase of this window would be beneficial? However it is worth implementing it. As I mentioned above, maybe also to envisage the inclusion of in situ surface measurements in the system that may enrich the satellite retrievals in terms of peaks (high temporal resolution in situ data) for some of these rapid source point events. Moreover, in general, the inclusion of in situ measurements may help to correct some biases in the satellite retrievals.

We agree that given the short-window used here high temporal resolution in-situ observations with low errors would improve results for specific hotspots nearby those available observations. On a global scale we expect this impact to be small; however, if further developments to the system were to focus on specific hotspots with in-situ observations then those would be included in any future work.

Overall, the methodology is sound and quite well described and especially the results are well presented and discussed. Moreover, the manuscript is well written and easy to read. After the clarifications of both the two issues highlighted above and some suggestions in my specific comments below, I would recommend the manuscript for publication in ACP

**Specific comments**

**Abstract**

Lines 17-18: …, *but without accurate prior uncertainty information, were not well quantified.'* Accurate? This is a bit misleading. This would mean that you have compared this uncertainty to a reference value. Maybe use as in the text: ' .. with appropriate prior uncertainty or your best prior information

We agree and have updated the text with your suggestions.

Lines 18-20: As already mentioned in my general comments, I do not understand why you performed the inversions only part of these years (i.e., 2019 and 2020). The comparison of the posterior estimates to the prior ones for the same period is fine, but the numbers for each of these years may be uncertain owing to the missing months.

We agree and have tried to add clarity that the values given represent the flux over a 6 month period and not an annual flux. For ease of reading we have kept the values as CH4 per year but have tried to emphasise the point that these are the fluxes for 6 months not a full year. We hope this is now clear enough.

1. **Introduction**

Lines 26-27: '*Changes in atmospheric chemistry, not investigated here, may have contributed to the observed growth in 2020*': How? The decrease of OH? Please elaborate.

We discuss these changes in more detail further down in the text but agree here we can also mention more specifically OH changes, this has now been included.

Line 80: '*…. at a high spatial and temporal resolution*' by at both high spatial and temporal resolutions ?

Updated thank you.

2. **Methods**

Lines 88-90: '*These were performed from January to June of 2019 and January to September of 2020..*'. I have not understood why the other months of these two years are not considered. Please elaborate

We have now included the justification being computational cost.

Line 133: Why do you consider the flux of the model LPJ-WHyme (page 96) and then you use the uncertainty from WetCHARTs. It is possible to consider both the mean estimate of CH4 emissions from WetCHARTs and the uncertainty as you do here

This is a good point, ideally we would have used the uncertainty and prior from the same source however the operational CAMS system has been well evaluated using the LPJ fluxes but an ensemble of fluxes is not available from the system. This is indeed a slight inconsistency with our assumption and we have now noted this in the text.

Lines 149-151: '*TROPOMI uncertainties provided as part of the CH4 product were applied within the minimisation routine and averaging kernels were use*d.' Give a reference of this data. May be the reference like Otto Hasekamp et al., (2017) [https://sentinel.esa.int/documents/247904/2476257/Sentinel-5P-TROPOMI-ATBD-Methane-retrieval] or an updated version?

Thank you, this has now been added.

Line 150: '*Additional CH4 observations from the …*' by Additional XCH4 observations from the …

3. **Results**

Lines 158: '… most of countries...' by most of the countries in the world.. ?

Updated, thank you.

Line 170: '*When evaluating XCH4 concentrations simulated with optimised emissions, the all-site average lowest standard error (6.8 ppb), absolute mean bias (7.52 ppb) and highest R-value (0.74) was found for the mapped prior error described in section 2.2.2.*'. Please give the range of R-value instead of only the highest R-value, otherwise give the lowest R-value.

This has now been updated.

**3.2. Global estimates.**

In this paragraph, I have not understood why you performed the inversions only for some months of the years 2019 and 2020. You compared your numbers to those of other studies in literature. Your numbers might include some additional uncertainties due the lack of these months. Moreover, the way you sample your optimised fluxes can introduce additional uncertainty in the numbers. In fact, what can be the contribution of the pixels discarded to the total numbers? I do not question the way you sampled your data, just be aware that doing so can introduce additional uncertainty and this needs to be quantified/mentioned in the paper. In fact, a rough estimation of the uncertainty due to your sampling method can be estimated by considering all the pixels to generate the numbers and compare them to those derived from your sampling method. I do not want to repeat these above remarks in other parts of the paper, hence please consider them and address them in the other parts of the paper when relevant.

We agree that reporting annual emissions was unclear and have adjusted this to report only monthly emissions. Given the reported values are for monthly averages, and assuming little variability within a month, we assume a subset of dates where observations are available within a month to be representative of the entire month. We agree this might introduce additional uncertainties and this is now updated earlier in the text. We have checked this assumption and whilst it does not always hold true, it is typically robust and the variability within a month is small.

**3.3. Emission estimates for Regions and Point Sources**

Line 205: '*To filter posterior estimates which provided little or no updated information we omitted daily grid cells associated with poor observation constraints (see supplement figure 1).*' Again, how much these pixels may contribute to the numbers? This remark is valid for the other parts of the study.

We accept this may introduce uncertainty and have updated the text accordingly.

Lines 226-227: '*While it is difficult to diagnose the cause of the difference in posterior estimates, one possibility is the larger prior uncertainty used in Zhang et al. (2020)*'. Yes, but not only the prior information can explain this. In fact, the inverse modelling system, as it is built, with a good observational coverage it may be possible to infer the fluxes from the surface whatever the prior information. Hence, here both prior information and observations

(here about 50% of the data are used and maybe no information in some pivotal areas) are meaningful to explain the deficiencies.

We agree and have added this in along with other possibilities regarding the period of study.

**3.3.4 Point source emissions- ….** Bracket after Australia to be deleted

Updated.

**3.5. CH4 emissions during the COVID-19 period**

Page 390: '*These 390 reduced emissions were likely caused by large scale droughts which occurred in early 2020 (Marengo et al., 2021'*. The droughts events may have decreased the water table?

Yes, we think this might be the case

**Conclusion**

Last paragraph: lines 459-464

Please adapt the text to the CH4 inverse modelling system. Here, you may mention only the new developments of the CO2 inverse modelling system that are relevant for the CH4 inversion system

This has now been updated, thank you.

**Tables**

OK

**Figures**

Figure 2:  C). The title of y-axis should be '% Change in Emissions  100.*(Posterior-Prior)/Prior

Updated.

**References**

Overall fine, only at page 16 move Courtier et al. 1994 after Cheewaphongphan et al. 2019

Updated.

**Supplement**

Figure 1: Quality flags. Please specify (>50% or larger ?)

Updated

---

## Referee Report (RR1)

**Overview:**
The manuscript "Quantification of methane emissions from hotspots and during COVID-19 using a global atmospheric inversion" by McNorton et al. describes the results of a high-resolution atmospheric inversion of methane ($CH_4$) emissions during 2019 and 2020. There is focus of many individual case studies of various scales, and investigation into the effect of the COVID-19 pandemic on global and regional emissions of $CH_4$.

The manuscript is much-improved since the previous iteration. Whilst I remain a little skeptical about the assumption that we should expect constant growth in emissions between 2020 and 2019 if the slowdown had not occurred, I do think that the authors have now made their reasoning more explicit and all assumptions and caveats are clearly documented.

Subject to very few small corrections, I am happy to recommend that this manuscript may be published.

**Minor corrections:**

Line 45: Rephrase – as written it sounds like the potential 2 ppb growth from OH variation is being contrasted against the 15.6 ppb observed global growth in 2020. However, it would be more sensible to compare it against the difference between the 2020 and 2019 growth rates. In this case, 2 ppb from OH might have accounted for around 35% of the increase in the growth rate between the two years (5.7 ppb), which is not insubstantial.

Line 48: No comma needed

Line 137: sector=specific -> sector-specific